# Alzheimer's disease linked Aβ42 exerts product feedback inhibition on γ-secretase impairing downstream cell signaling

Katarzyna Marta Zoltowska[1], Utpal Das[2], Sam Lismont[1], Thomas Enzlein[1,3], Masato Maesako[4], Mei CQ Houser[4], Maria Luisa Franco[5], Burcu Özcan[1], Diana Gomes Moreira[1], Dmitry Karachentsev[2], Ann Becker[2], Carsten Hopf[3,6,7], Marçal Vilar[5], Oksana Berezovska[4], William Mobley[2]*, Lucía Chávez-Gutiérrez[1]*

[1]VIB-KU Leuven Center for Brain & Disease Research, Leuven, Belgium; [2]Department of Neurosciences, University of California San Diego, La Jolla, United States; [3]Center for Mass Spectrometry and Optical Spectroscopy (CeMOS), Mannheim University of Applied Sciences, Mannheim, Germany; [4]Department of Neurology, Massachusetts General Hospital/Harvard Medical School, Charlestown, United States; [5]Molecular Basis of Neurodegeneration Unit, Instituto de Biomedicina de Valencia, Valencia, Spain; [6]Medical Faculty, Heidelberg University, Heidelberg, Germany; [7]Mannheim Center for Translational Neuroscience (MCTN), Medical Faculty Mannheim, Heidelberg University, Heidelberg, Germany

**\*For correspondence:**
wmobley@ucsd.edu (WM);
lucia.chavezgutierrez@kuleuven.
vib.be (LC-G)

**Abstract** Amyloid β (Aβ) peptides accumulating in the brain are proposed to trigger Alzheimer's disease (AD). However, molecular cascades underlying their toxicity are poorly defined. Here, we explored a novel hypothesis for Aβ42 toxicity that arises from its proven affinity for γ-secretases. We hypothesized that the reported increases in Aβ42, particularly in the endolysosomal compartment, promote the establishment of a product feedback inhibitory mechanism on γ-secretases, and thereby impair downstream signaling events. We conducted kinetic analyses of γ-secretase activity in cell-free systems in the presence of Aβ, as well as cell-based and *ex vivo* assays in neuronal cell lines, neurons, and brain synaptosomes to assess the impact of Aβ on γ-secretases. We show that human Aβ42 peptides, but neither murine Aβ42 nor human Aβ17–42 (p3), inhibit γ-secretases and trigger accumulation of unprocessed substrates in neurons, including C-terminal fragments (CTFs) of APP, p75, and pan-cadherin. Moreover, Aβ42 treatment dysregulated cellular homeostasis, as shown by the induction of p75-dependent neuronal death in two distinct cellular systems. Our findings raise the possibility that pathological elevations in Aβ42 contribute to cellular toxicity via the γ-secretase inhibition, and provide a novel conceptual framework to address Aβ toxicity in the context of γ-secretase-dependent homeostatic signaling.

## eLife assessment

In this manuscript, the authors tested the hypothesis that Aβ42 toxicity arises from its proven affinity for γ-secretases. The authors provide **useful** findings, showing **convincingly** that human Abeta42 inhibits gamma-secretase activity. The data will be of interest to all scientists working on neurodegenerative diseases.

## Introduction

Γ-secretases are ubiquitously expressed intramembrane proteases best known for their pathogenic roles in Alzheimer's Disease (AD) (*Chávez-Gutiérrez and Szaruga, 2020*). Aberrant processing of the amyloid precursor protein (APP) by γ-secretases leads to the production of longer, aggregation-prone Aβ peptides that contribute to neurodegeneration (*Selkoe and Hardy, 2016*).

In addition, γ-secretases process many other membrane proteins, including NOTCH, ERB-B2 receptor tyrosine kinase 4 (ERBB4), N-cadherin (NCAD), and p75 neurotrophin receptor (p75-NTR) (*Haapasalo and Kovacs, 2011*; *Güner and Lichtenthaler, 2020*). The processing of multiple substrates links their activity to a broad range of downstream signaling pathways (*Jurisch-Yaksi et al., 2013*; *Carroll and Li, 2016*), including those critical for neuronal function. It is noteworthy that treatments with γ-secretase inhibitors caused cognitive worsening in AD patients (*Doody et al., 2013*), while full genetic inhibition of these enzymes in the adult mouse brain led to neurodegenerative phenotypes (*Acx et al., 2017*; *Wines-Samuelson et al., 2010*; *Tabuchi et al., 2009*; *Saura et al., 2004*; *Bi et al., 2021*). The underlying mechanisms by which the deficits in γ-secretase activity impair neuronal function are yet to be defined.

Γ-secretase activity is exerted by a family of highly homologous multimeric proteases composed of presenilin (PSEN1 or PSEN2), nicastrin (NCSTN), anterior pharynx defective 1 (APH1A or B), and presenilin enhancer 2 (PEN2) subunits. The proteolytic activities of these complexes are promoted by the low pH of the endosomal and lysosomal compartments, wherein the amyloidogenic processing of APP occurs (*Maesako et al., 2022*). In the amyloidogenic pathway, the proteolytic processing of APP by β-secretase (BACE) releases a soluble APP ectodomain and generates a membrane-bound C-terminal fragment (β-CTF or $APP_{C99}$) (*Vassar et al., 1999*). $APP_{C99}$ is then sequentially processed within the membrane by γ-secretase complexes (*Figure 1A*; *Takami et al., 2009*; *Bolduc et al., 2016*; *Chávez-Gutiérrez et al., 2012*; *Qi-Takahara et al., 2005*; *Funamoto et al., 2004*). An initial endopeptidase (ε-) cut releases the APP intracellular domain (AICD) into the cytosol and generates a *de novo* substrate (either Aβ49 or Aβ48 peptide) that undergoes successive γ-cleavages until a shortened Aβ peptide can be released into the luminal or extracellular environment. The efficiency of the sequential cleavage mechanism (i.e. processivity) determines the length of Aβ (37–43 amino acid long peptides), which in turn influences the aggregation and neurotoxic properties of the peptides produced (*Selkoe and Hardy, 2016*; *Kakuda et al., 2017*; *Fu et al., 2017*). In the non-amyloidogenic pathway APP is cleaved by α- and γ-secretases to generate a spectrum of p3 peptides, which lack the first 1–16 amino acids of Aβ (*Figure 1A*). Despite their relatively high hydrophobicity and aggregation-prone behavior, the p3 peptides are not linked to AD pathogenesis (*Kuhn and Raskatov, 2020*; *Lichtenthaler, 2011*; *Tambini et al., 2020*). In fact, mutations that promote the amyloidogenic processing of APP are associated with AD (*Mullan et al., 1992*; *Pagnon de la Vega et al., 2021*), whereas those that favor the alternative, non-amyloidogenic pathway protect against the disease (*Tambini et al., 2020*; *Jansen et al., 2019*).

In familial AD (FAD), mutations in PSENs or in APP/Aβ (i.e. changes effecting both the enzyme and its substrate) promote the generation of longer Aβ peptides, including Aβ42, but also Aβ43 (*Veugelen et al., 2016*; *Fernandez et al., 2014*; *Szaruga et al., 2017*; *Devkota et al., 2021*; *Kretner et al., 2016*; *Petit et al., 2022*). These peptides accumulate in the brain and are a hallmark of AD, in both familial and sporadic forms. In the latter (SAD), the accumulation of longer, aggregation-prone peptides results from their inefficient clearance (*Mawuenyega et al., 2010*; *Liu et al., 2023*). Irrespective of the mechanism, the accumulation of Aβ in the brain begins decades before the onset of clinical symptoms and is proposed to trigger toxic cascades via poorly understood mechanisms (*Selkoe and Hardy, 2016*).

Aggregation of Aβ42 into soluble toxic oligomers has been proposed as a prerequisite for its toxicity (*Brinkmalm et al., 2019*; *Walsh et al., 2002*; *Hong et al., 2018*; *Wang et al., 2017*). However, given the broad spectrum of brain-derived Aβ peptides and their assemblies, it seems plausible that Aβ contributes via a number of mechanisms to AD pathogenesis. Of note, increasing evidence suggests that in addition to Aβ (*Selkoe and Hardy, 2016*), its precursor, the 99 amino acid C-terminal fragment (i.e. $APP_{C99}$) (*Lauritzen et al., 2019*) also mediates pathogenic mechanisms (*Xu et al., 2016*; *Kwart et al., 2019*; *Kim et al., 2016*).

Herein, we explored a novel hypothesis for Aβ42 toxicity that arises from its proven affinity for γ-secretases. We hypothesized that pathological increases in Aβ levels in the AD brain lead to the

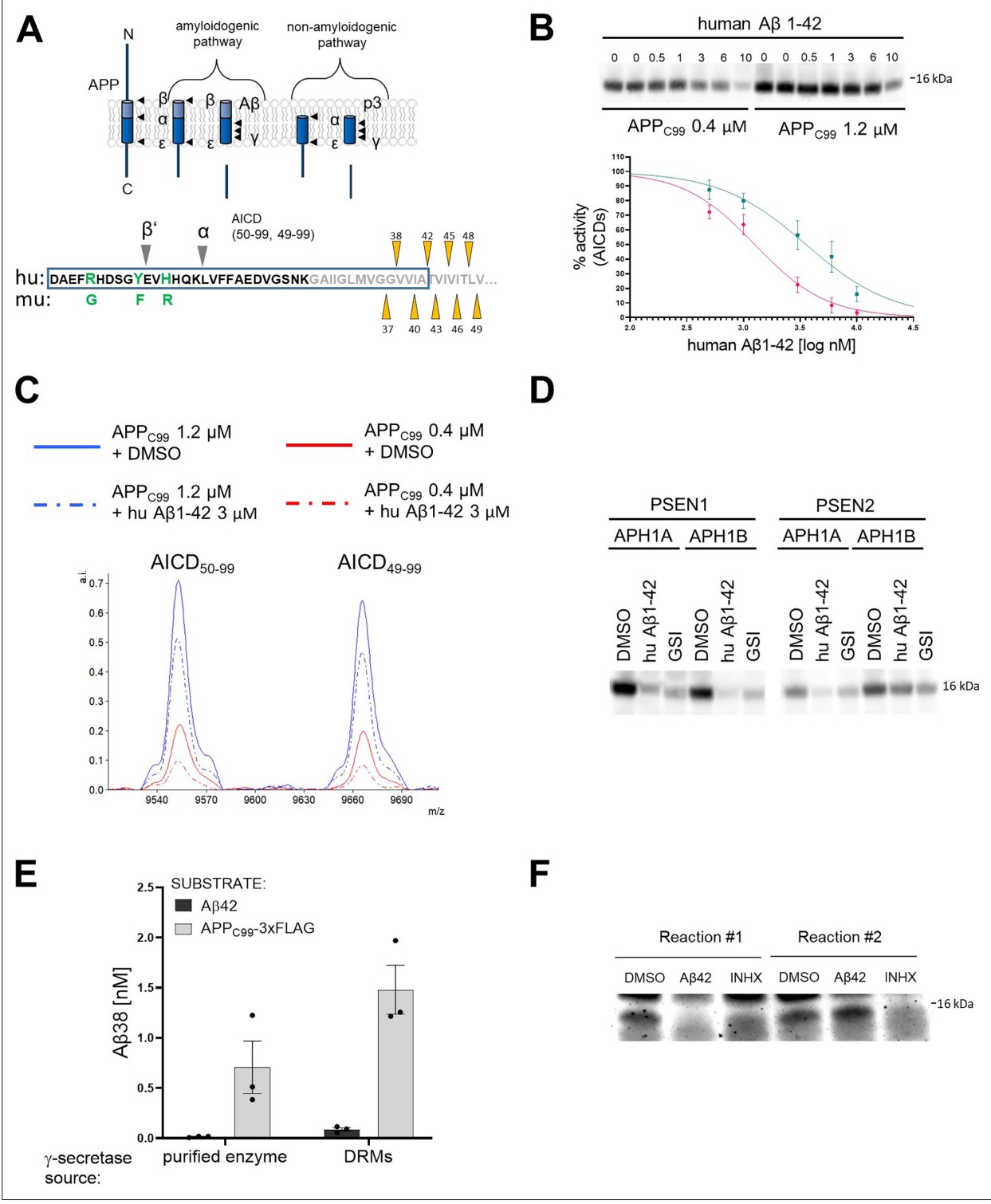

**Figure 1.** Human Aβ42 peptide inhibits γ-secretase-mediated proteolysis of APP_C99. (**A**) The scheme depicts the γ-secretase-mediated cleavage of amyloid precursor protein (APP), leading to the generation of amyloid β (Aβ) and p3 peptides. The N-terminal sequence of APP_C99 /Aβ is shown in the lower panel. The differences in the amino acid sequence of human (hu) vs murine (mu) Aβ peptides and the positions of β'- and α-cleavages (that precede the generation of Aβ11–42 and p3 17–42 peptides, respectively) are indicated. The transmembrane domain is labeled in grey and the sequence of Aβ42 is presented within a rectangle. The initial γ-secretase endopeptidase cut may occur at one of two different positions on APP, generating two different *de novo* substrates that are further processed by carboxypeptidase-like cleavages as follows: Aβ49→Aβ46→Aβ43→Aβ40→Aβ37 or Aβ48 →Aβ45→Aβ42→Aβ38. Aβ38, Aβ40, and Aβ42 peptides are the major products under physiological conditions. The triangles mark the sequential

*Figure 1 continued on next page*

*Figure 1 continued*

cleavage positions. (**B**) The western blot presents APP intracellular domain (AICD) products generated *de novo* in detergent-based γ-secretase activity assays using APP$_{C99}$-3xFLAG at 0.4 µM or 1.2 µM as substrate. To test the inhibitory properties of human Aβ1–42, the peptide was added to the activity assays at concentrations ranging from 0.5 to 10 µM. DMSO at 2.5% was used as a vehicle control. The graphs present the quantification of the western blot bands corresponding to AICDs. The pink and green lines correspond to 0.4 µM and 1.2 µM substrate concentrations, respectively. The data are normalized to the AICD levels generated in the DMSO conditions, considered as 100%. The data are presented as mean ± SEM, n=6–16. (**C**) Mass spectrometry-based analysis of *de novo* generated AICD levels in *in vitro* cell-free γ-secretase activity assays containing 3 µM human Aβ1–42 or vehicle (0.6% DMSO) indicate that human Aβ1–42 inhibits both product lines. (**D**) The western blot presents *de novo* generated AICDs in detergent-based γ-secretase activity assays using 0.4 µM APP$_{C99}$-3xFLAG as substrate and wild-type γ-secretases composed of different presenilin (PSEN) (1 or 2) and anterior pharynx defective 1 (APH1) (A or B) subunits in the presence of vehicle, human Aβ1–42 at 3 µM or GSI (InhX) at 10 µM concentration. (**E**) The graph presents ELISA quantification of Aβ1–38 peptides generated in detergent- or detergent-resistant membranes (DRM)-based γ-secretase activity assays using either human Aβ1–42 at 10 µM or APP$_{C99}$-3xFLAG at 1.5 µM as substrates. The data are presented as mean ± SEM, n=3. (**F**) The western blot presents *de novo* generated AICDs in detergent-based γ-secretase activity assays using 0.4 µM APP$_{C99}$-3xFLAG as a substrate and γ-secretase immobilized on sepharose beads. During the first round, the activity assay was supplemented with 3 µM Aβ1–42, 10 µM GSI or DMSO vehicle. After this first round, the γ-secretase-beads were washed to remove the peptide and inhibitor, respectively, fresh substrate was added, and the reaction proceeded for the second round. The analysis demonstrates the reversibility of the Aβ1–42-mediated inhibition.

The online version of this article includes the following source data and figure supplement(s) for figure 1:

**Source data 1.** The unedited blots and figures with the uncropped blots with the relevant bands clearly labeled.

**Figure supplement 1.** Cell-free γ-secretase activity assay.

**Figure supplement 1—source data 1.** The unedited blots and figures with the uncropped blots with the relevant bands clearly labeled.

establishment of a product feedback inhibitory mechanism, wherein Aβ1–42 competes with membrane-associated substrates for γ-secretase processing. This hypothesis is supported by our reported observations that Aβs with low affinity for γ-secretase, when present at relatively high concentrations, can compete with the longer, higher affinity APP$_{C99}$ substrate for binding and processing by the enzyme (*Szaruga et al., 2017*). This hypothetical inhibitory feedback would result in the accumulation of unprocessed γ-secretase substrates and reduced levels of their soluble intracellular domain (ICD) products, possibly disrupting downstream signaling cascades.

To test the hypothesis, we investigated whether Aβ peptides can exert inhibition on γ-secretase. Our kinetic analyses demonstrate that human Aβ1–42 inhibits γ-secretase-mediated processing of APP$_{C99}$ and other substrates. Strikingly, neither murine Aβ1–42 nor human p3 (17–42 amino acids in Aβ) peptides exerted inhibition under similar conditions. We also show that human Aβ1–42-mediated inhibition of γ-secretase activity results in the accumulation of unprocessed CTFs of APP, p75, and pan-cadherins. To evaluate the impact of the Aβ-driven inhibition on cellular signaling, we analyzed p75-dependent activation of caspase 3 in basal forebrain cholinergic neurons (BFCNs) and PC12 cells. These analyses demonstrate that, as seen for γ-secretase inhibitors (*Franco et al., 2021*), Aβ1–42 potentiates this marker of apoptosis. Our findings thus point to an entirely novel and selective role for the Aβ42 peptide, and raise the intriguing possibility that compromised γ-secretase activity against the CTFs of APP and/or other substrates contributes to the pathogenesis of AD.

## Results

### Aβ1-42 inhibits γ-secretase-mediated proteolysis of APP$_{C99}$

We have shown that Aβ peptides displaying a relatively low affinity for γ-secretase, when present at high concentrations, can compete with the higher affinity APP$_{C99}$ substrate for binding to the enzyme (*Szaruga et al., 2017*). Based on this observation, we hypothesized that an increase in the concentration of Aβ42 may promote the establishment of an inhibitory mechanism that involves the formation of 'non-productive' enzyme-Aβ42 complexes. The Aβ42-mediated inhibition would result in the accumulation of unprocessed γ-secretase substrates and reduced production of their intracellular domains, both of which may contribute to dysregulated downstream signaling.

As a first step, we investigated the effects of human Aβ1–42 (from now on referred to as Aβ42) on the processing of APP$_{C99}$ in well-controlled kinetic analyses (*Figure 1B*; *Szaruga et al., 2017*). We incubated purified (wild-type) γ-secretase enzyme with purified APP$_{C99}$-3xFLAG substrate at the K$_M$ or saturating concentrations (0.4 µM and 1.2 µM, respectively) in the presence of human Aβ42 peptides at concentrations ranging from 0.5 µM to 10 µM. We then analyzed the *de novo* generation

of AICD-3xFLAG by quantitative western blotting (*Figure 1—figure supplement 1A*). Methanol:chloroform extraction was performed to remove the excess of unprocessed substrates, the high levels of which could preclude the quantitative analysis of ICDs. The analysis demonstrated that human Aβ42 inhibited γ-secretase-mediated proteolysis of $APP_{C99}$ in a dose-dependent manner. The more pronounced peptide-driven inhibition at the lower substrate concentration ($IC_{50}$=1.3 µM vs 3.6 µM for 0.4 and 1.2 µM $APP_{C99}$, respectively) is consistent with a mechanism in which Aβ42 competes with the substrate for the enzyme. In addition, mass spectrometry analyses of the proteolytic reactions showed that Aβ42 partially inhibited the generation of both AICD product types ($AICD_{50-99}$ and $AICD_{49-99}$) (*Figure 1C*), indicating that Aβ42 inhibits both γ-secretase product lines. Next, we tested whether human Aβ42 exerted inhibition on all members of the γ-secretase family – i.e., irrespective of the type of PSEN (1 vs 2) and APH1 (A vs B) subunits (*Figure 1D*). Quantitative western blotting analysis revealed a marked inhibition of total AICD production by all types of γ-secretases in the presence of 3 µM human Aβ42. These findings support a competitive mechanism wherein low-affinity substrates (acting also as products) are able to re-associate with the protease and inhibit the processing of transmembrane substrates when present at relatively high concentrations.

To gain further insights, we investigated γ-secretase mediated processing of Aβ42 to Aβ38 under the conditions used to examine $APP_{C99}$, using the latter as a positive control (*Figure 1E*). Despite the use of relatively high concentrations of Aβ42 (10 µM), this peptide was not converted into Aβ38. In contrast, proteolytic reactions using $APP_{C99}$ (1.5 µM) resulted in the generation of Aβ38 (0.5–1 nM). We also tested whether Aβ42 served as a substrate in conditions that mimic a native-like environment, i.e., detergent-resistant membranes (DRMs) (*Figure 1E*; *Szaruga et al., 2017*; *Matsumura et al., 2014*). As in the detergent conditions, Aβ42 was barely converted into Aβ38. We note that γ-secretase processes Aβ43 into Aβ40 under similar conditions, even when this peptide was added at much lower concentrations (0.5–1 µM) (*Szaruga et al., 2017*). Taken together, these observations indicate that exogenous Aβ42 does interact with γ-secretases but, unlike Aβ43, does not act as a substrate (at least under these conditions), supporting the notion that Aβ42-driven inhibition of γ-secretases is mediated via the formation of non-productive enzyme-substrate (E-S) like complexes. However, a scenario wherein Aβ42 interacts with $APP_{C99}$ to reduce the amount of free $APP_{C99}$ substrate available for the enzymatic cleavage is not excluded by these data.

We also investigated whether the inhibitory effects of Aβ42 on γ-secretase were reversible. To this end, we conjugated purified γ-secretase complexes to beads using a high-affinity anti-NCSTN nanobody and incubated the enzyme-conjugated beads with 0.4 µM $APP_{C99}$, in the absence or presence of 3 µM Aβ42, for 40 min at 37 °C. Note that this concentration of peptide substantially inhibited AICD generation (*Figure 1B*). As a control, 10 µM γ-secretase inhibitor X (GSI, Inh X) was included. After the incubation, we collected the supernatants, washed the beads in assay buffer, and re-incubated them with 0.4 µM $APP_{C99}$ for 40 min at 37 °C. Analysis of the levels of the *de novo* generated AICD products in the supernatant fractions collected before (reaction 1) and after washes (reaction 2) indicated that Aβ42 inhibition of γ-secretase is fully reversible (*Figure 1F*). Collectively, our analyses support a model wherein Aβ42 forms a non-productive E-S-like complex with γ-secretase and its binding is reversible.

## The N- and C-termini of Aβ play key roles in the inhibition of γ-secretase activity

We then investigated the structure-function relationships relevant to the Aβ42-driven inhibitory mechanism. The effects of mouse/rat (murine) Aβ42 and N-terminally truncated human Aβx-42 (11–42 and 17–42) peptides on γ-secretase activity were examined in cell-free assays using peptide concentrations ranging from 0.5 µM to 10 µM (*Figure 2A–C*). Quantification of the *de novo* AICD product levels showed that murine Aβ42 maximally inhibited γ-secretase activity by ~20% (*Figure 2A*). As three amino acids in the N-terminal domain (R5G, Y10F and H13R) differentiate human and murine Aβ1–42 peptides (*Figure 1A*), the differences in the inhibition thus defined the N-terminal domain of Aβ as contributing to the inhibitory mechanism. It is noteworthy that similar to human Aβ1–42, murine Aβ1–42 was not processed to Aβ1–38 (*Figure 1—figure supplement 1B*). The analyses of other naturally occurring N-terminally truncated Aβx-42 peptides, generated by β-secretase (alternative) cleavage at the position 11 or by α-secretase cut at the position 17 in the Aβ sequence, showed that the truncated peptides exhibited reduced inhibitory potencies relative to Aβ42. The $IC_{50}$ values for Aβ11–42 were reduced 1.79- and 1.31-fold ($K_M$ and saturating substrate concentrations, respectively),

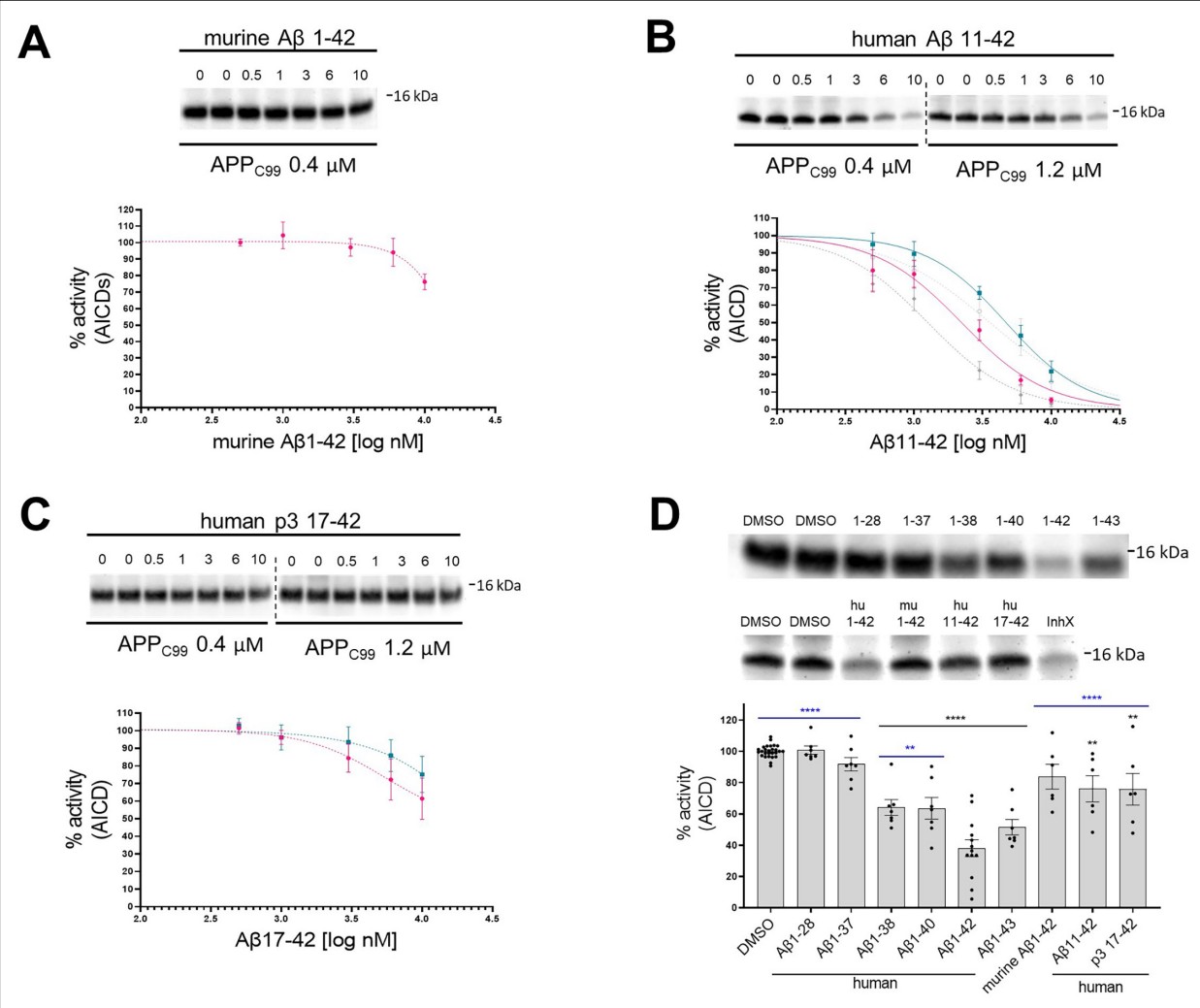

**Figure 2.** N- and C-terminus of amyloid β (Aβ) influence the inhibitory properties of the peptide. (**A, B, C**) The western blots present *de novo* generated APP intracellular domains (AICDs) in detergent-based γ-secretase activity assays using purified protease and APP$_{C99}$-3xFLAG at 0.4 μM or 1.2 μM as a substrate. To test the inhibitory properties of (**A**) murine Aβ1–42, (**B**) human Aβ11–42, and (**C**) human p3 17–42, the peptides were added to the assays at concentrations ranging from 0.5 to 10 μM. DMSO at 2.5% was used as a vehicle. The graphs present the quantification of the western blot bands for AICDs. The pink and green lines correspond to 0.4 μM and 1.2 μM substrate concentrations, respectively. The grey dotted lines in the plot B present the curves recorded for human Aβ1–42 (from *Figure 1*, plot B) for comparison. The data are normalized to the AICD levels generated in the DMSO conditions, considered as 100%, and presented as mean ± SEM, n=3–8. (**D**) Detergent-based γ-secretase activity assays using purified protease and APP$_{C99}$-3xFLAG at 0.4 μM concentration were supplemented with different Aβ peptides at 1 μM concentration or DMSO. *De novo* generated AICDs were quantified by western blotting. The data are shown as mean ± SEM, n=6–27. The statistics were calculated using one-way ANOVA and multiple comparisons Dunnett's test, with DMSO (black) or Aβ1–42 (blue) set as references. **p<0.01, ****p<0.0001.

The online version of this article includes the following source data for figure 2:

**Source data 1.** The unedited blots and figures with the uncropped blots with the relevant bands clearly labeled.

relative to Aβ42 (*Figure 2B*, *Supplementary file 1a*), while the larger N-terminal truncation (of residues 1–16) even further reduced the inhibitory effect to the level seen with murine Aβ42 (*Figure 2C*). Collectively, these data assign a defining role to the N-terminal region of Aβ in the inhibition of γ-secretase activity.

We next tested whether longer Aβ peptides (>Aβ42), which form more stable interactions with the protease (*Szaruga et al., 2017*), also inhibit γ-secretase activity. In addition, we investigated whether shortening of the C-terminus diminishes the inhibitory properties. To this end, we evaluated a series of naturally occurring Aβ1-x peptides, ranging from 37 to 43 amino acids in length, at 1 μM final concentration for their effect on γ-secretase proteolysis of APP$_{C99}$ (*Figure 2D*). In these experiments,

the reduction in DMSO (vehicle) concentration to 0.2% was accompanied by ~1.7 fold enhanced Aβ-mediated inhibition, relative to the conditions shown in *Figure 1B*. We speculate that increased Aβ potency under these conditions is explained by a modest but measurable reduction in the proteolysis by the higher concentration of DMSO used in *Figure 1* assays (*Shelton et al., 2009*). Relative to Aβ42, shorter Aβ species exerted progressively less or no inhibition on the *de novo* generation of AICD. It is noteworthy that peptides longer than Aβ42 serve as substrates for γ-secretases under these conditions, and their shortening potentially converts them into less potent inhibitory species (*Szaruga et al., 2017*). Taken together, these data strengthen the conclusion that human Aβ42 inhibits γ-secretases and indicate that both Aβ C- and N-termini modulate the inhibitory mechanism.

## Aβ42 treatment leads to the accumulation of APP C-terminal fragments in neuronal cell lines and human neurons

We reasoned that human Aβ42-driven inhibition of γ-secretase activity would lead to the accumulation of unprocessed γ-secretase substrates in cells (e.g. APP-CTFs in the case of APP) as well as reduced generation of products (Aβ peptides in the case of APP). To test these possibilities, we treated human neuroblastoma SH-SY5Y, rat pheochromocytoma PC12, and human neural progenitor cells (ReNcell VM) with recombinant human Aβ42 or p3 17–42 peptides at 1 µM or 2.5 µM final concentrations, and analyzed the levels of endogenous APP-CTFs and full-length APP (APP-FL) by western blotting (*Figure 3A–C*). Treatment with GSI (InhX) at 2 µM was included as a positive control. Human Aβ42 treatment increased the APP-CTF/FL ratio, a read-out of γ-secretase activity. In contrast, treatment with p3 peptide had no effect. The increments in the APP-CTF/FL ratio suggested that Aβ42 (partially) inhibits the global γ-secretase activity. To further investigate this, we measured the direct products of the γ-secretase-mediated proteolysis of APP. Since the detection of the endogenous Aβ products via standard ELISA methods was precluded by the presence of exogenous human Aβ42 (treatment), we used an N-terminally tagged version of APP$_{C99}$ and quantified the amount of total secreted Aβ, which is a proxy for the global γ-secretase activity. Briefly, we overexpressed human APP$_{C99}$ N-terminally tagged with the short 11 amino acid long HiBiT tag in human embryonic kidney (HEK) cells, treated these cultures with human Aβ42 or p3 17–42 peptides at 1 µM, or DAPT (GSI) at 10 µM, and determined total HiBiT-Aβ levels in conditioned media (CM). DAPT fully inhibits γ-secretase at the concentration used, and hence the values measured in DAPT-treated conditions were used for the background subtraction. We found a ~50% reduction in luminescence signal, directly linked to HiBiT-Aβ levels, in CM of cells treated with human Aβ42 and no effect of p3 peptide treatment, relative to the DMSO control (*Figure 3D*). The observed reduction in the total Aβ products is consistent with the partial inhibition of γ-secretase by Aβ42.

In addition, to controlling for the overall cellular toxicity of Aβ42, we performed two cell toxicity assays that rely on different biological principles (*Figure 3E*). In the first one, we quantified the activity of lactate dehydrogenase (LDH) released by cells into the conditioned medium upon plasma membrane damage (*Figure 3F*). In the second, we detected cellular ATP as a reporter of viability. There was no significant change in these measures in response to Aβ42 (1 µM), p3 17–42 (1 µM) or GSI (2 µM) (*Figure 3G*).

We next asked if the effects of Aβ42 would be also registered in wild-type human neurons. We thus examined neurons derived from induced pluripotent stem cells (iPSC) and treated with human or murine Aβ42, p3 17–42, or human Aβ40 (all at 2.5 µM). Only treatment with human Aβ42 resulted in a significant increase of APP-CTFs (*Figure 3H*), supporting the inhibitory role of Aβ42.

## Selective accumulation of Aβ42 in cells leads to increased levels of APP C-terminal fragments

Next, we examined the effects of a broader spectrum of Aβ peptides, presenting different N- and C-termini, on APP-CTF levels in SH-SY5Y cells. In addition, we tested murine Aβ42, which failed to inhibit γ-secretase activity in cell-free assays (*Figure 4A*). The analysis confirmed a significant ~2.5 fold increase in the APP-CTF/APP-FL ratio in cells treated with human Aβ42 but, intriguingly, none of the other investigated peptides significantly changed APP-CTF levels.

These observations provided evidence that Aβ42 differs from other tested peptides in one or more features that are critical for γ-secretase inhibition in cells. Previous studies have shown that selective cellular uptake of human Aβ42, relative to Aβ40, leading to its concentration in the acidic

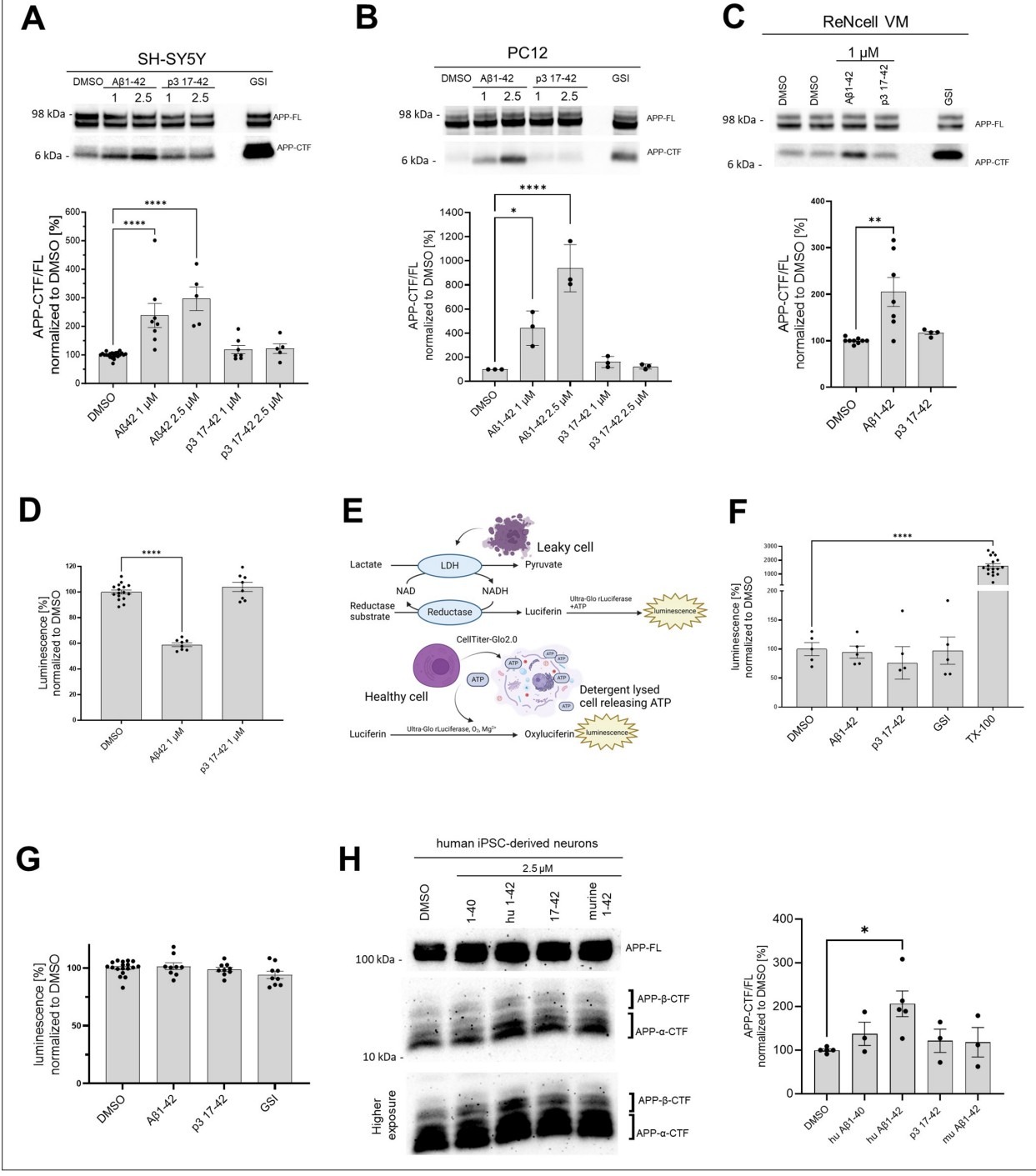

**Figure 3.** Human Aβ42 leads to the accumulation of APP-CTFs. (**A, B, C, H**) The western blots present full-length amyloid precursor protein (APP) (APP-FL) and APP C-terminal fragments (APP-CTFs) detected in (**A**) SH-SY5Y, (**B**) PC12, (**C**) ReNcell VM human neural progenitor cells and (**H**) induced pluripotent stem cell-derived human neurons treated for 24 h with respective peptides at indicated concentrations or vehicle (DMSO). The ratio between the APP-CTF and APP-FL levels was calculated from the integrated density of the corresponding western blot bands. The data are shown as mean ± SEM, n=3–23. The statistics were calculated using one-way ANOVA and multiple comparison Dunnett's test, with DMSO set as a reference. *p<0.05, **p<0.01, ****p<0.0001. (**D**) The amount of HiBiT-Aβ peptides was measured in the conditioned medium collected from human embryonic kidney (HEK) cell line stably expressing HiBiT-APP C99 treated with DMSO, Aβ1–42 (1 µM) or p3 17–42 (1 µM). The data are shown as mean ± SEM, n=8–16. The statistics were calculated using one-way ANOVA and multiple comparisons Dunnett's test, with DMSO, set as a reference. ****p<0.0001. (**E**) The scheme presents the principles of the lactate dehydrogenase (LDH)-based and ATP-based cytotoxicity assays. (**F**) Cytotoxicity of the treatments was analyzed in SH-SY5Y cells. The cells were treated with DMSO, Aβ1–42 (1 µM), p3 17–42 (1 µM), or GSI (InhX, 2 µM) for 24 h, conditioned medium

*Figure 3 continued on next page*

*Figure 3 continued*

collected and subjected to the measurement of lactate dehydrogenase (LDH) activity using luminescence-based assay. TX-100 was used as a positive control expected to lead to 100% cell death. The data are shown as mean ± SEM, n=5–17. The statistics were calculated using one-way ANOVA and multiple comparisons Dunnett's test, with DMSO set as a reference, ****p<0.0001. TX-100 led to a marked increase in the luminescent signal, while no significant toxicity of the other treatments was detected. (**G**) An ATP-based cell viability assay was used to determine the cytotoxicity of the treatments. We analyzed SH-SY5Y cells treated with DMSO, Aβ1–42 (1 µM), p3 17–42 (1 µM), or GSI (Inh X, 2 µM) for 24 h. The data are shown as mean ± SEM, n=9–18. The statistics were calculated using one-way ANOVA and multiple comparisons Dunnett's test, with DMSO set as a reference. No significant toxicity of the treatments was detected.

The online version of this article includes the following source data for figure 3:

**Source data 1.** The unedited blots and figures with the uncropped blots with the relevant bands clearly labeled.

endolysosomal network (ELN), promotes peptide aggregation into soluble toxic Aβ species (*Hu et al., 2009*; *Su and Chang, 2001*; *Schützmann et al., 2021*; *Liu et al., 2010*; *Esbjörner et al., 2014*; *Friedrich et al., 2010*). We reasoned that since this compartment is the main locus of γ-secretase activity (*Maesako et al., 2020*), the concentration of Aβ42 in ELN may not only promote its aggregation but also facilitate its inhibitory actions on γ-secretases. To investigate whether the selective accumulation of Aβ42 in ELN, relative to the other tested peptides, could explain the differential (cellular) inhibitory profiles (*Figure 4B*), we treated PC12 cells with human Aβ40, Aβ42, Aβ43, or p3 17–42, or murine Aβ42 peptides at 1 µM final concentration for 24 h, and examined their intracellular pools by immunostaining with an anti-Aβ/p3 antibody, 4G8 (epitope: 17–23 aa). We found that, unlike other Aβ or p3 peptides, human Aβ42 accumulated in cells. The pattern of the accumulation appeared distinct, punctate, and largely perinuclear, suggestive of its presence in the endolysosomal compartment. The data are consistent with a model in which intracellular accumulation of Aβ42, due to a selective cellular uptake or reduced peptide degradation, distinguishes it from the other peptides studied and explains its apparently unique inhibitory properties in the cellular context.

### Human Aβ42 inhibits endogenous γ-secretase activity in neurons

We next asked whether the accumulation of APP-CTFs in cells stems from direct inhibition of γ-secretases. To answer this question, we applied an established cell-based γ-secretase activity assay to test *in situ* the protease activity in primary mouse neurons (*Maesako et al., 2020*; *Houser et al., 2020*; *Figure 5A*). This assay uses an APP$_{C99}$-based fluorescent substrate (the C99 Y-T biosensor) to probe γ-secretase activity in living cells. This biosensor comprises APP$_{C99}$ fused at the C-terminus with YPet, followed by a linker, Turquoise-GL, and a membrane-anchoring domain to stabilize the probe in the membrane. The cleavage of the APP$_{C99}$-based probe by endogenous γ-secretases extends the distance between YPet and Turquoise-GL, and therefore results in a decrease in the efficiency of Förster resonance energy transfer (FRET) for this pair. The ratiometric nature of the reporter, and its independence of α- and β-secretase activity and cellular degradation mechanisms, allow quantitative analysis of γ-secretase activity *in situ* in living cells.

We treated mouse primary neurons with human Aβ42, p3, GSI (1 µM DAPT), or vehicle (DMSO) for 24 h and performed FRET analysis (*Figure 5B–C*). While p3 peptides did not affect the FRET signal, significant increases in FRET efficiency (i.e. increased proximity between YPet and Turquoise-GL) in cells treated with human Aβ42 or GSI, relative to vehicle-treated cultures, indicated increments in the levels of the full-length (uncleaved) fluorescent substrate, and thus reduced γ-secretase activity. These results provided quantitative evidence in a cellular context for the inhibition of endogenous γ-secretases by human Aβ42, but not p3 17–42 peptides.

### Aβ42-mediated γ-secretase inhibition is the major mechanism contributing to APP-CTF accumulation in cells

Our data support a direct effect of Aβ42 peptides on the activity of γ-secretase in well-controlled, cell-free systems and in cell-based assays. Although the cell-free and FRET-based systems are independent of α- and β-secretase activity and cellular degradation pathways, we elected to further pursue the possibility that changes in the activity of these enzymes or degradation of CTFs could contribute to the observed increase in APP-CTFs in cellular systems. We investigated the impact of Aβ42 on

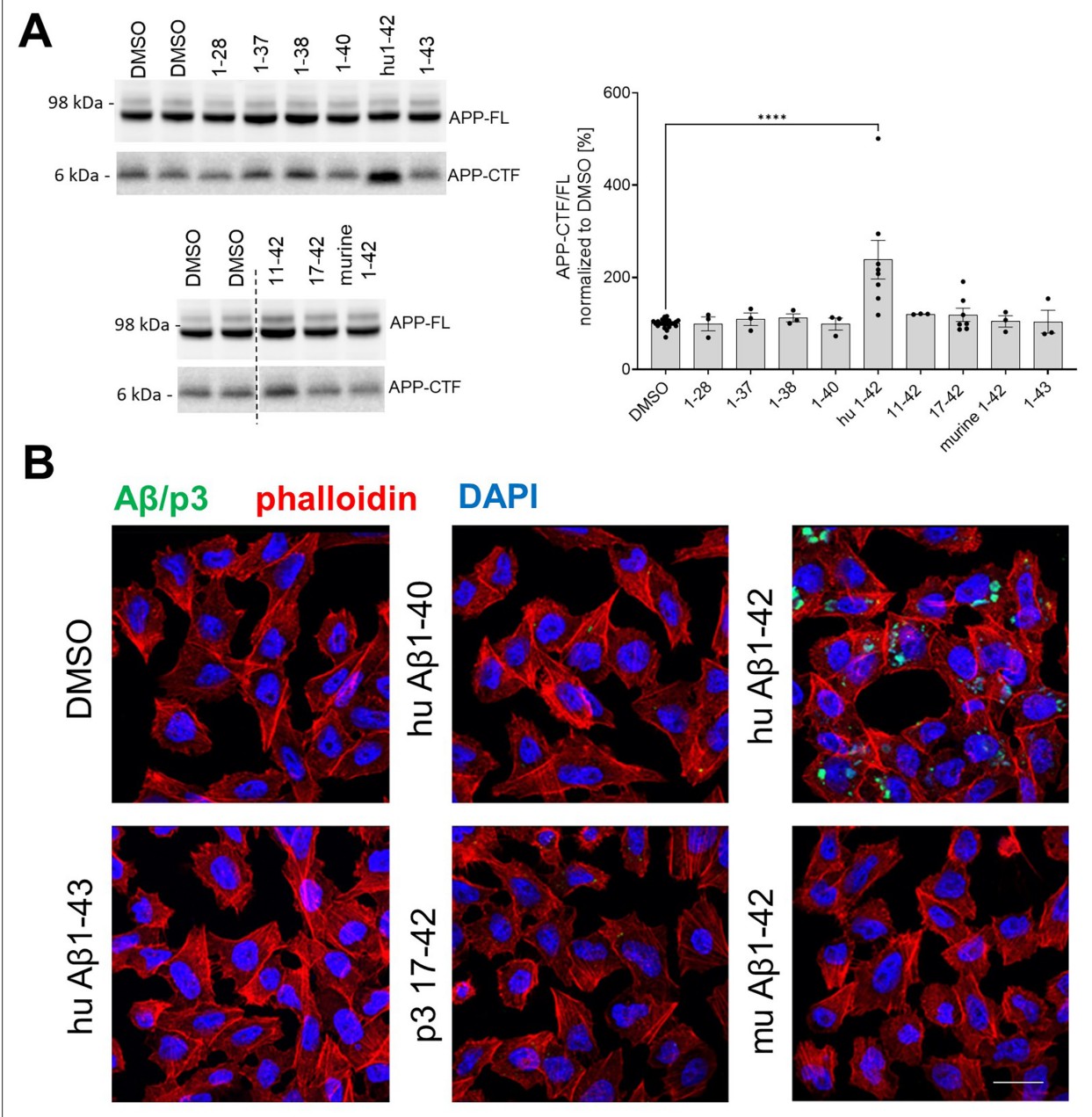

**Figure 4.** Human Aβ42 selectively accumulates in the cells and inhibits γ-secretase-mediated proteolysis. (**A**) APP-FL and APP-CTF levels in SH-SY5Y cells treated for 24 h with a series of amyloid β (Aβ) peptides at 1 μM concentration are shown. The APP-CTF/FL ratio was calculated from the integrated density of the corresponding western blot bands. The data are shown as mean ± SEM, n=3–23. The statistics were calculated using one-way ANOVA and multiple comparisons Dunnett's test, with DMSO set as a reference. ****p<0.0001. (**B**) PC12 cells were treated with respective Aβ or p3 peptides for 24 h and stained with anti-Aβ antibody (clone 4G8) followed by anti-mouse Alexa Fluor Plus 488 conjugated secondary antibody, Alexa Fluor Plus 555 conjugated phalloidin and nuclear stain DAPI. Scale bar: 20 μm.

The online version of this article includes the following source data for figure 4:

**Source data 1.** The unedited blots and figures with the uncropped blots with the relevant bands clearly labeled.

α-secretase (ADAM10) and β-secretase (BACE1) levels, alterations in which could result in increased APP-CTF levels, by analyzing the expression of ADAM10 and BACE1 in total cell lysates from SH-SY5Y cells treated with Aβ40 (1 μM), Aβ42 (1 μM), p3 17–42 (1 μM), GSI (InhX, 2 μM), or vehicle DMSO control (*Figure 6—figure supplement 1*). We found no evidence of increased either α-secretase or β-secretase levels. In addition, we explored possible differences in the activities of these secretases

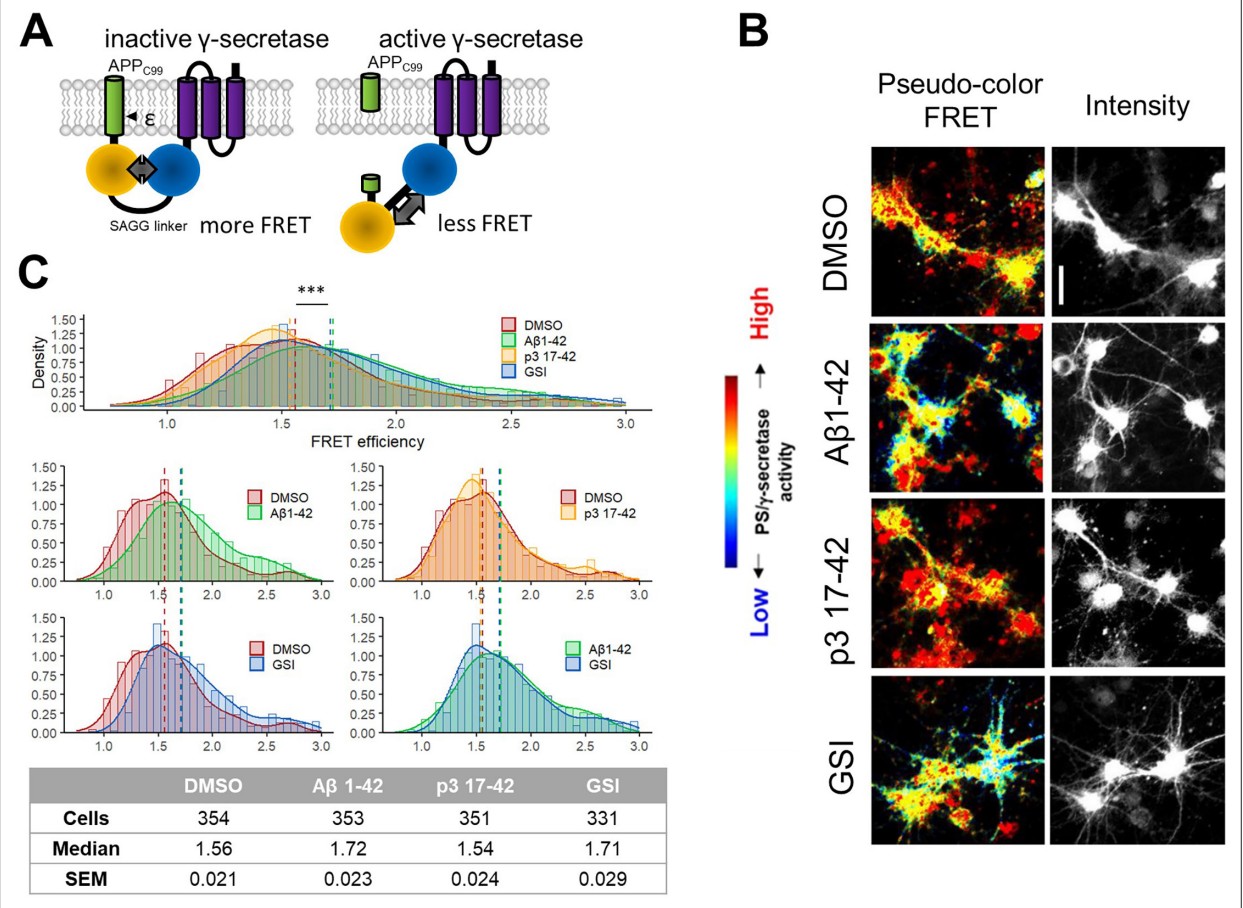

**Figure 5.** Human Aβ42 inhibits endogenous γ-secretase activity in neurons. (**A**) The scheme presents the Förster resonance energy transfer (FRET)-based probe allowing monitoring of γ-secretase activity *in situ* in living cells. (**B, C**) Spectral FRET analysis of γ-secretase activity in mouse primary neurons using a C99 Y-T probe is shown. The cells were treated with the indicated peptides/compounds at 1 µM concentration for 24 h. Γ-secretase-mediated proteolysis results in an increase in the distance between two fluorophores incorporated in the probe (YPet and Turquoise-GL). The increase in the distance translates to the reduced FRET efficiency, quantified by the YPet/Turquoise-GL fluorescence ratio. The distribution of recorded FRET efficiency, inversely correlating with γ-secretase activity, is shown in the density plots, n=4 (n=331–354). Medians are shown as dashed lines. Optimal bin number was determined using the Freedman-Diaconis rule. The statistics were calculated using the Kruskal-Wallis test and multiple comparison Dunn test. Significant differences (***p<0.001) were recorded for DMSO vs Aβ1–42, p3 17–42 vs Aβ1–42, DMSO vs GSI, and p3 17–42 vs GSI. Scale bar: 20 µm.

by examining total soluble APP (sAPP) in a conditioned medium collected from SH-SY5Y cells treated under the same conditions. Our analysis showed a reduction in total sAPP levels in the conditioned medium collected from cells treated with Aβ42. This finding is inconsistent with an Aβ42-mediated effect that would increase APP-CTF levels as a result of activating α-secretase or β-secretase activity. Further studies are required to define whether the decrease in total sAPP is a consequence of sheddase inhibition or relates to an effect of Aβ42 on the rate of recycling and release of sAPP fragments.

We also evaluated potential γ-secretase independent changes in the degradation of APP-CTF by assessing the impact of Aβ42 peptides on γ-secretase independent APP-CTF half-life using the well-established cycloheximide (CHX)-based assay. CHX inhibits protein synthesis by blocking translation elongation (*Schneider-Poetsch et al., 2010*). In these experiments, we pre-treated SH-SY5Y cells with either GSI (DAPT, 10 µM) or GSI + Aβ42 (1 µM) for 24 h. Importantly, the GSI treatment allowed for complete γ-secretase inhibition in both conditions, resulting in the maximum accumulation of APP-CTFs that can possibly be achieved by blocking γ-secretase processing. We then added CHX (50 µg/ml) while maintaining GSI and vehicle versus GSI plus Aβ42. In addition, we treated with bafilomycin A1 (200 nM), which de-acidifies lysosomes and compromises degradation pathways, as a positive control (*Figure 6A*). We collected cells thereafter at 0 h, 1 h, 2.5 h and 5 h.

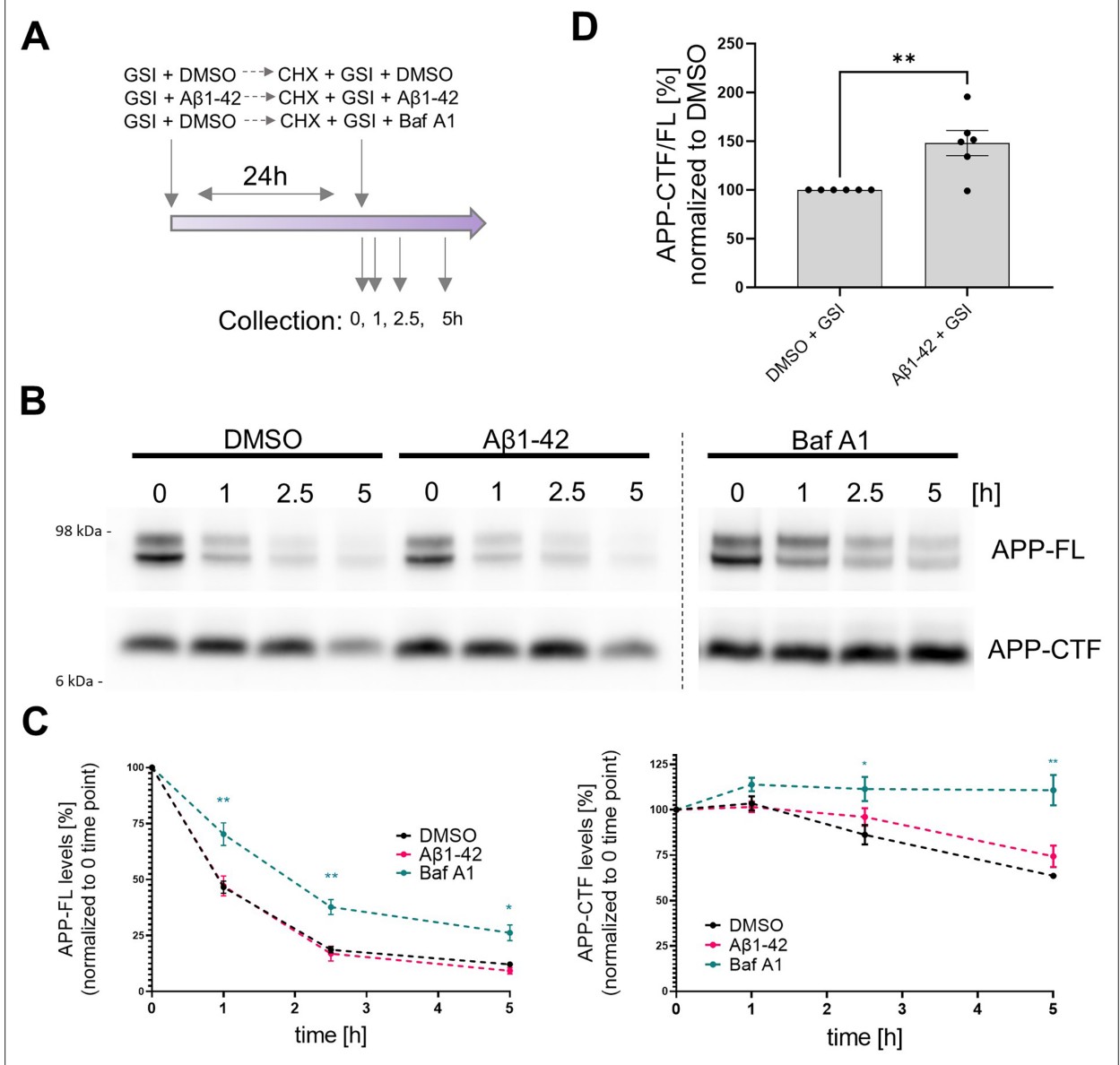

**Figure 6.** Contribution of Aβ42-mediated inhibition of γ-secretase and general degradation mechanisms to APP-CTF accumulation. (**A**) The scheme presents the experimental design of the cycloheximide (CHX)-based assay evaluating APP-FL and APP-CTF stability. (**B**) Western blot shows APP-FL and APP-CTF levels in SH-SY5Y cells at 0, 1, 2.5, and 5 h collection points defined in the scheme (**A**). (**C**) The integrated densities of the bands corresponding to APP-FL and APP-CTF were quantified and plotted relatively to the time point zero. The data are presented as mean ± SEM, n=6. Statistics were calculated using two-way ANOVA, followed by multiple comparisons Dunnett's test. *p<0.05, **p<0.01. (**D**) Quantification of APP-CTF/ FL ratio at the zero time point is shown. The data are presented as mean ± SEM, n=6. Statistics were calculated using unpaired Student's t-test, **p<0.01.

The online version of this article includes the following source data and figure supplement(s) for figure 6:

**Source data 1.** The unedited blots and figures with the uncropped blots with the relevant bands clearly labeled.

**Figure supplement 1.** Sheddase activation is not behind the APP-CTF accumulation in cells.

**Figure supplement 1—source data 1.** The unedited blots and figures with the uncropped blots with the relevant bands clearly labeled.

Quantitative analysis of the levels of both APP-CTFs and APP-FL over the 5 h time-course showed that bafilomycin A1 treatment markedly prolonged the half-life of both proteins but did not reveal significant differences in their levels between GSI versus GSI plus Aβ42-treated cells (*Figure 6B–C*). The lack of a significant effects of Aβ42 on the levels of APP-CTFs under these conditions (full γ-secretase inhibition) suggests that the peptide does not induce major defects in cellular degradation pathways, at least at this concentration. However, we noted an increased APP-CTF/FL ratio in GSI + Aβ42

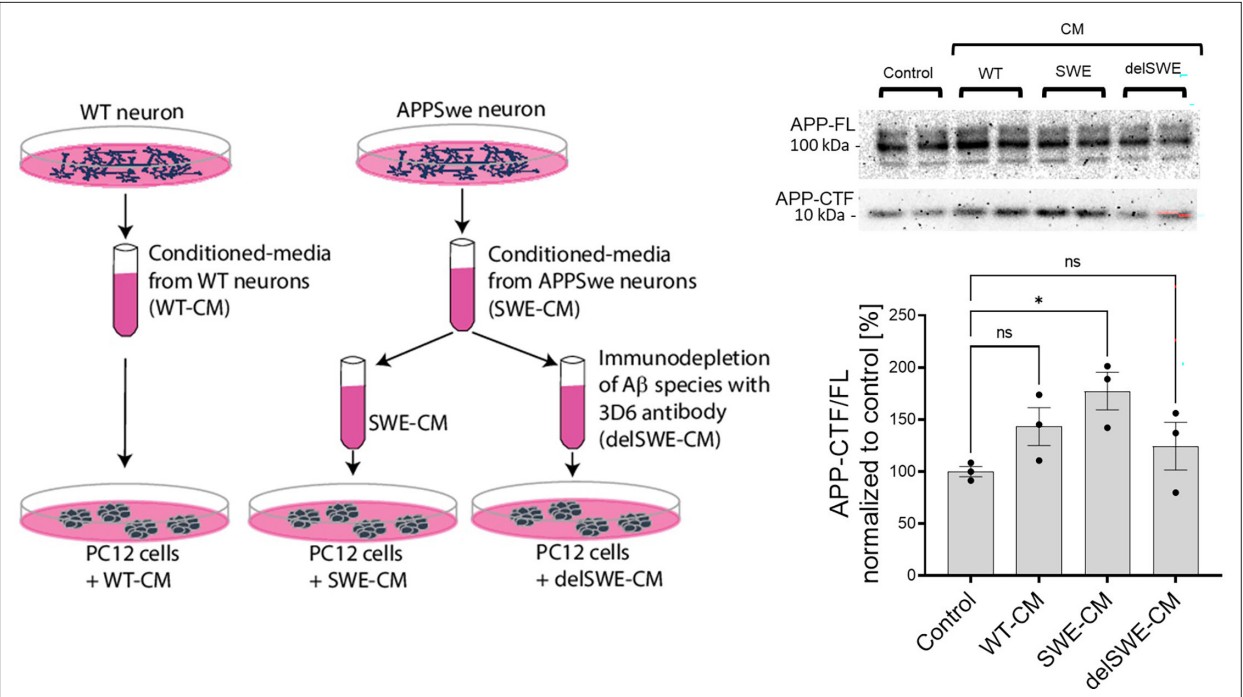

**Figure 7.** Human Aβ42 leads to the accumulation of APP-CTFs. Scheme depicts the experimental design testing the impact of biologically-derived Aβ on the proteolysis of amyloid precursor protein (APP). Conditioned media were collected from fully differentiated wild-type (WT) human neurons (WT-CM) and human neurons expressing APP with Swedish mutation (SWE-CM). A portion of SWE-CM was subjected to amyloid β (Aβ) immunodepletion using the anti-Aβ antibody 3D6. WT, SWE, and Aβ immunodepleted (delSWE-CM) conditioned medias (CMs) were added to the PC12 cells for 24 h to analyze APP processing. As a reference, control cells treated with base media were analyzed. Representative western blots present the analysis of total cellular proteins from four cell culture sets treated with base media (control), WT-CM, SWE-CM, and delSWE-CM, respectively. The ratio between APP-CTFs and APP-FL was calculated from the integrated density of the corresponding western blot bands. The data are shown as mean ± SEM, n=3. The statistics were calculated using one-way ANOVA and multiple comparisons Dunnett's test, with a control set as a reference, *p<0.05.

The online version of this article includes the following source data for figure 7:

**Source data 1.** The unedited blots and figures with the uncropped blots with the relevant bands clearly labeled.

treated cells vs GSI + DMSO treated ones at time zero (*Figure 6D*), suggesting that the Aβ42-induced increase in APP-CTFs is mediated by both inhibition of γ-secretase and one or more additional mechanisms. Nevertheless, the greater relative accumulation of APP-CTFs in the absence (138%) (*Figure 3A*) versus the presence (48%) of the GSI (*Figure 6D*) is evidence that Aβ42-mediated γ-secretase inhibition plays a more prominent role.

## Aβ peptides derived from biological sources trigger APP C-terminal fragment accumulation

We next asked whether Aβ from disease-relevant biological sources would recapitulate the effects of recombinant Aβ42. For these studies, we used a culture medium conditioned by human iPSC-derived neurons expressing APP wild-type or the KM670/671 NL (SWE) variant. The APP SWE mutation increases total Aβ production by promoting the amyloidogenic processing of APP. ELISA-based quantification of Aβ content estimated a total concentration in the low nanomolar range (0.5–1 nM). Conditioned medium (WT or SWE) was applied to PC12 cells either directly or after Aβ depletion (delSWE) using the anti-APP 3D6 (epitope Aβ 1–5) (*Figure 7*). After a 24 h incubation at 37°C, the APP-CTF/FL ratio was measured. Cells incubated in control, unconditioned media were used as a reference control. We observed significant increments in the APP-CTF/FL ratio in cells treated with SWE-conditioned media, relative to control cells. Aβ immunodepletion from the SWE medium (delSWE-CM) lowered the APP-CTF/FL ratio to the levels observed in the control cells. These data provide evidence that Aβ peptides derived from biological sources induce the accumulation of APP-CTFs even

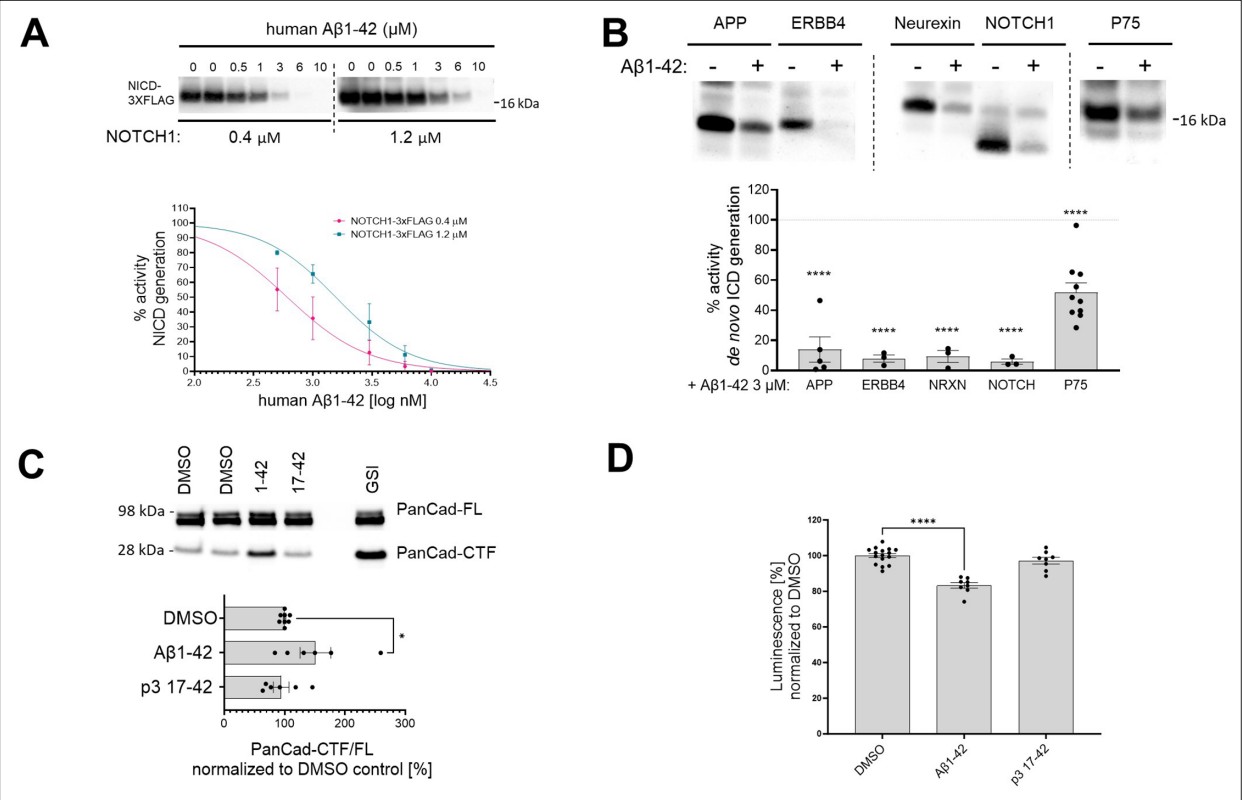

**Figure 8.** Human Aβ1–42 peptides inhibit proteolysis of multiple γ-secretase substrates. (**A**) The western blot presents *de novo* generated NICDs in detergent-based γ-secretase activity assays, using NOTCH1-3xFLAG at 0.4 µM and 1.2 µM as a substrate, supplemented with human Aβ1–42 peptides at concentrations ranging from 0.5 to 10 µM. The graphs present the quantification of the western blot bands for NICDs. The pink and green lines correspond to 0.4 µM and 1.2 µM substrate concentrations, respectively. The data are normalized to the NICD levels generated in the DMSO conditions, considered as 100%, and presented as mean ± SEM, n=3–5. (**B**) Analysis of the *de novo* intracellular domain (ICD) generation in cell-free detergent-based γ-secretase activity assays is shown. The graph presents the quantification of the western blots. The data are shown as mean ± SEM, n=3–18. The statistics were calculated using one-way ANOVA and multiple comparisons of predefined columns, with the Šidák correction test, with respective DMSO-supplemented reactions set as a reference, ****p<0.0001. (**C**) PanCad-FL and PanCad-CTF levels in ReNcell VM cells treated for 24 h with human Aβ1–42 peptides at 1 µM or GSI (InhX) at 2 µM concentration were quantified by western blotting. The PanCad-CTF/FL ratio was calculated from the integrated density of the corresponding bands. The data are presented as mean ± SEM, n=6–8. The statistics were calculated using one-way ANOVA and multiple comparisons Dunnett's test, with DMSO set as a reference. *p<0.05. (**D**) The graph presents the quantification of the HiBiT-Aβ like peptide levels in conditioned media collected from HEK cell line stably expressing the HiBiT-NOTCH1 based substrate and treated with DMSO, Aβ1–42 (1 µM), or p3 17–42 (1 µM). The data are shown as mean ± SEM, n=8–16. The statistics were calculated using one-way ANOVA and multiple comparisons Dunnett's test, with DMSO set as a reference. ****p<0.0001.

The online version of this article includes the following source data and figure supplement(s) for figure 8:

**Source data 1.** The unedited blots and figures with the uncropped blots with the relevant bands clearly labeled.

**Figure supplement 1.** Mass spectrometry analysis confirms the inhibitory action of human Aβ1–42 peptides.

when present at low nM concentrations, and thus point to this biological source of Aβ as being far more potent than the recombinant Aβ42.

## Human Aβ42 peptides inhibit proteolysis of other γ-secretase substrates, beyond APP-CTFs

The Aβ42-mediated inhibition of endogenous γ-secretase activity in cells raises the possibility that Aβ42 would inhibit the processing of γ-secretase substrates beyond APP-CTFs. To address this, we analyzed the effects of human Aβ42 on the processing of a purified NOTCH1-based substrate in cell-free conditions (*Figure 8A*). We incubated purified γ-secretase and the NOTCH1-based substrate (at 0.4 µM or 1.2 µM concentrations) in the presence of increasing amounts of human Aβ1–42 (ranging from 0.5 µM to 10 µM). Quantification of the *de novo* generated NICD-3xFLAG showed a

dose-dependent inhibition of γ-secretase proteolysis of NOTCH1 by Aβ42. As for APP$_{C99}$, the derived IC$_{50}$ values showed that the degree of the inhibition depended on substrate concentration, consistent with a competitive mechanism (*Supplementary file 1a*). We extended this analysis by assessing the γ-secretase-mediated proteolysis of other substrates (ERBB4-, neurexin-, and p75-based) in the presence of 3 μM human Aβ42 (*Figure 8B*). Quantification of the respective *de novo* generated ICDs revealed that Aβ42 reduced γ-secretase proteolysis of each of these substrates. Mass spectrometry-based analyses of the respective ICDs confirmed the inhibition (*Figure 8—figure supplement 1*).

We next investigated the effects of human Aβ42 on γ-secretase-mediated processing of pan-cadherin in ReNcell VM cells. Cleavage of cadherins by α-secretase generates a membrane-bound CTF, which is a direct substrate of γ-secretases (*Uemura et al., 2006*). As before, we used the PanCad-CTF/FL ratio to estimate the efficiency of the γ-secretase-mediated proteolytic processing (*Figure 8C*). Treatment with human Aβ42 and InhX, but not p3, resulted in significant increases in the PanCad-CTF/FL ratio, demonstrating that Aβ42 inhibits the processing of several different γ-secretase substrates. These results suggest that this peptide can inhibit the proteolysis of γ-secretases substrates in general.

Finally, we measured the direct N-terminal products generated by γ-secretase proteolysis from a HiBiT-tagged NOTCH1-based substrate, an estimate of the global γ-secretase activity. We quantified the Aβ-like peptides secreted by HEK 293 cells stably expressing this HiBiT-tagged substrate upon treatment with 1 μM Aβ1–42, p3 17–42 peptide or 10 μM DAPT (GSI) as control for background subtraction (*Figure 8D*). A~20% significant reduction in the amount of secreted N-terminal HiBiT-tagged peptides derived from the NOTCH1-based substrates in cells treated with Aβ1–42 supports the inhibitory action of Aβ1–42 on γ-secretase mediated proteolysis.

By demonstrating that Aβ42 inhibits γ-secretase proteolysis of substrates that are structurally distinct from APP-CTFs, these data indicate that Aβ42 inhibits enzyme activity independent of substrate structure. This renders much less likely the mechanistic possibility that inhibition is due to an interaction with the substrate.

## Human Aβ42-driven accumulation of p75-CTF induces markers of neuronal death

The p75 neurotrophin receptor plays a prominent role in neurotrophin signaling. Under certain conditions, its C-terminal fragment (p75-CTF, a γ-secretase substrate) modulates cell death (*Wong et al., 2022*; *Conroy and Coulson, 2022*). Previous studies have shown that accumulation of p75-CTF in basal forebrain cholinergic neurons (BFCNs), due to pharmacological inhibition of γ-secretases, triggers apoptosis in a TrkA activity-dependent manner (*Franco et al., 2021*). The proapoptotic action of p75-CTF is prevented by the expression of the TrkA receptor for nerve growth factor (NGF), which serves as a co-receptor with p75 (*Franco et al., 2021*). Similar observations have been made in PC12 cells, which model sympathetic neurons. In this case, γ-secretase inhibitors increased apoptosis in TrkA-deficient (PC12nnr5) (*Loeb et al., 1991*), but not in wild-type, TrkA-expressing cells. These findings provided an experimental system to investigate the effects of Aβ42-mediated inhibition on γ-secretase-mediated signaling in neurons and neuron-like cells.

We tested the impact of Aβ on p75 signaling in wild-type and PC12nnr5 cells. We exposed the cells to 1 μM human Aβ42, 1 μM p3, GSI (10 μM compound E), or vehicle (DMSO) for 3 days, and measured apoptosis via immunofluorescence staining for cleaved caspase 3 (*Figure 9A*). In addition, to assess γ-secretase proteolysis of p75, we measured the p75-CTF/FL ratio in the treated cultures (*Figure 9B*). Cleaved caspase 3 staining was increased in PC12nnr5 cells treated with GSI or human Aβ42, but not in those exposed to p3 or vehicle (DMSO). Of note, and consistent with our previous findings (*Franco et al., 2021*), none of the treatments elicited significant increases in cleaved caspase 3 in wild-type PC12 cells that express physiological levels of TrkA. In addition, analysis of the p75-CTF/FL ratio demonstrated higher p75-CTF levels in GSI and Aβ42 treated cells. These results, together with our previous observations, strongly suggest that Aβ42-mediated accumulation of p75-CTFs results in cell death signaling in the absence or near absence of TrkA expression. Our similar studies using BFCNs recapitulated the reported increased cell death under conditions in which both γ-secretase and TrkA activities were inhibited (CE and K252α inhibitor) (*Franco et al., 2021*), and showed that treatment with 1 μM human Aβ42 mimics the increase in cleaved caspase 3 driven by the GSI (*Figure 9C*). In conclusion, these results are evidence of a novel role for human Aβ42 peptide in the global inhibition of γ-secretase activity and dysregulation of cellular homeostasis.

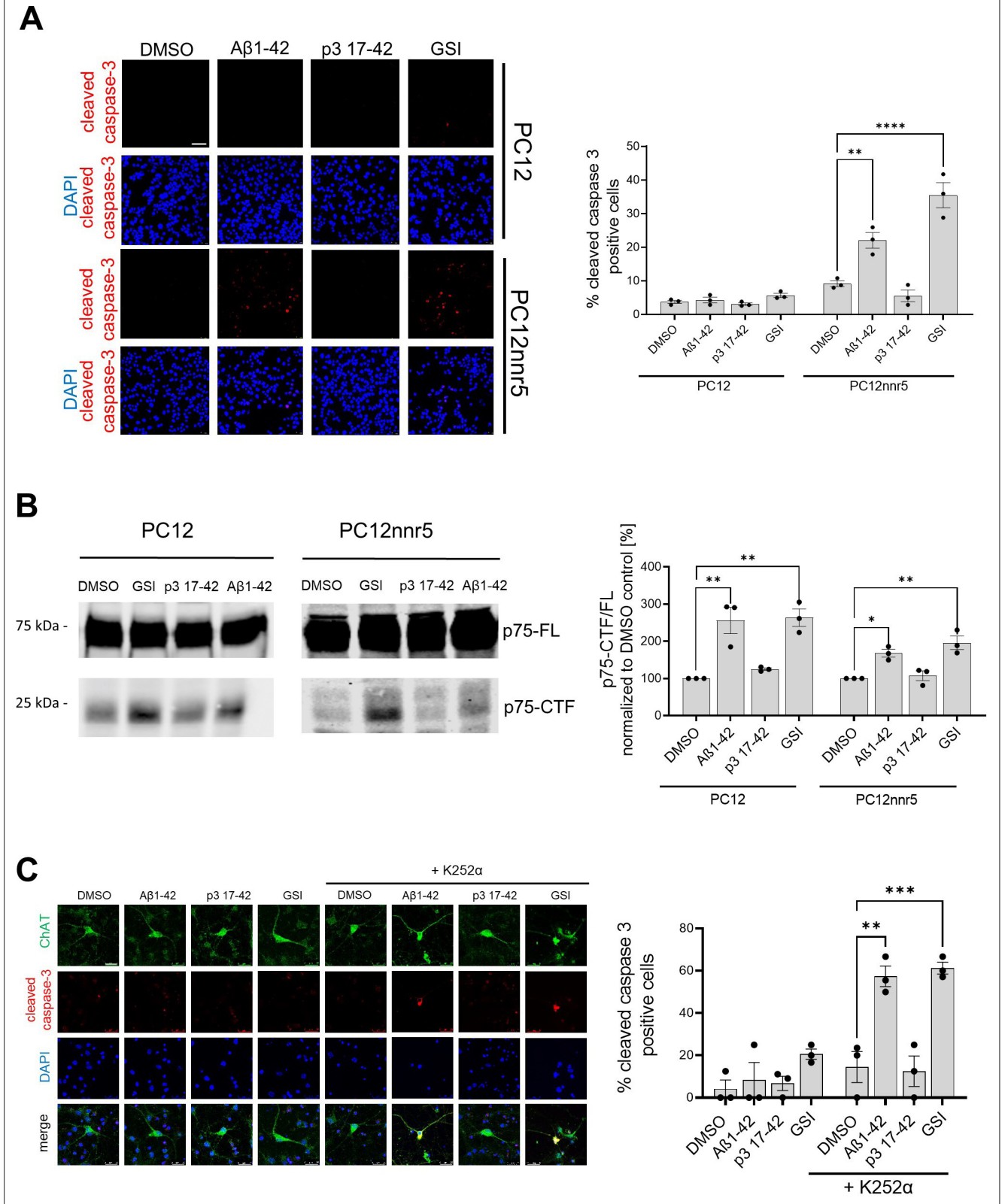

**Figure 9.** Human Aβ1–42 peptides compromises p75-mediated signaling. (**A**) PC12 wild-type or PC12 deficient for TrkA (PC12nnr5) were incubated with human Aβ1–42 or p3 17–42 peptides, γ-secretase (GSI) (compound E), or vehicle (DMSO) for 72 h. The images present immunocytochemical analyses of cleaved caspase 3, and the graph corresponding quantification of the percentage of cleaved caspase 3 positive cells. Scale bar 50 μm. The statistics were calculated within PC12 and PC12nnr5 groups using one-way ANOVA and multiple comparison Dunnett's test, with DMSO set as a reference. The

*Figure 9 continued on next page*

*Figure 9 continued*

data are presented as mean ± SEM, n=3. **p<0.01,****p<0.0001. (**B**) Representative western blot demonstrates the accumulation of p75-CTFs in the cells treated with human Aβ1–42 peptide or GSI. The data are presented as mean ± SEM, n=3. The statistics were calculated within PC12 and PC12nnr5 groups using one-way ANOVA and multiple comparison Dunnett's test, with DMSO set as a reference. *p<0.05, **p<0.01. (**C**) Mouse primary neurons were treated with human Aβ1–42 (1 µM), p3 17–42 (1 µM), GSI (compound E, 10 µM), or vehicle control, in the absence or presence of K252α inhibitor at 0.5 µM. Level of apoptosis in basal frontal cholinergic neurons (BFCNs) was analyzed by immunostaining for choline acetyltransferase (ChAT) and cleaved caspase 3. Representative images are shown, scale bar 25 µm. The graph presents the quantification of the percentage of cleaved caspase-3 positive cells among ChAT-positive cells. The data are presented as mean ± SEM, n=3. The statistics were calculated within -K252α and + K252α groups using one-way ANOVA and multiple comparison Dunnett's test, with DMSO set as a reference. **p<0.01, ***p<0.001.

The online version of this article includes the following source data for figure 9:

**Source data 1.** The unedited blots and figures with the uncropped blots with the relevant bands clearly labeled.

## APP-CTF levels are increased in Aβ42-treated synaptosomes derived from mouse brains

Since both the amyloidogenic processing of APP and the accumulation of Aβ occur at the synapse (*Pickett et al., 2016*; *Schedin-Weiss et al., 2016*), we reasoned that synaptosomes are potentially a cellular locus where elevated levels of Aβ could be acting on γ-secretase. We thus tested whether γ-secretase-mediated processing of APP at the synapse is inhibited by Aβ42 (*Figure 10A*). To this end, we prepared synaptosomes from wild-type mouse brains, treated these fractions with 2.5 µM Aβ42 or vehicle, and then measured APP-FL and APP-CTF levels by western blotting. An increased APP-CTF/FL ratio in Aβ42 treated samples, compared to vehicle control, provided evidence of γ-secretase inhibition.

We next investigated the levels of Aβ42 in synaptosomes derived from frontal cortices of post-mortem AD and age-matched non-demented (ND) control individuals (*Figure 10B*). Towards this, we prepared synaptosomes from frozen brain tissues using the Percoll gradient procedure (*Dunkley et al., 2008*; *Fonseca-Ornelas et al., 2021*). Intact synaptosomes were spun to obtain a pellet which was resuspended in the minimum amount of PBS, allowing us to estimate the volume containing the resuspended synaptosome sample. This is likely an overestimate of the actual synaptosome volume. Finally, synaptosomes were lysed in RIPA buffer and Aβ peptide concentrations were measured using ELISA (MSD). We observed that the concentration of Aβ42 in the synaptosomes from (end-stage) AD tissues was significantly higher (10.7 nM) than those isolated from non-demented tissues (0.7 nM), ***p<0.0005. These data provide evidence for accumulation at nM concentrations of endogenous Aβ42 in synaptosomes in end-stage AD brains. Given that we measured Aβ42 concentration in synaptosomes, we speculate that even higher concentrations of this peptide may be present in the endolysosome vesicle system, and therein inhibit the endogenous processing of APP-CTF at the synapse. Of note treatment of PC12 cells with conditioned medium containing even lower amounts of Aβ (low nanomolar range (0.5–1 nM)) resulted in the accumulation of APP-CTFs.

## Discussion

Compelling evidence supports that Aβ peptides trigger molecular and cellular cascades leading to neurodegeneration (*Scheltens et al., 2021*; *Knopman et al., 2021*). Here, we discovered and extensively characterized a previously unreported role for Aβ: inhibition of the activity of γ-secretase complexes (*Figure 10C and D*). The recognition that Aβ peptides, despite lower affinities, can compete with APP-CTFs for binding to γ-secretase, when present at relatively high concentrations (*Szaruga et al., 2017*), led us to propose a mechanism that connects increases in the Aβ42 peptide with the inhibition of γ-secretases and dysregulation of signaling cascades relevant to neuronal physiology. In this regard, we note that the inhibition of γ-secretase in the adult mouse brain leads to age-dependent neurodegenerative phenotypes (*Acx et al., 2017*; *Wines-Samuelson et al., 2010*; *Tabuchi et al., 2009*; *Saura et al., 2004*) by poorly understood but APP-independent mechanisms (*Acx et al., 2017*), whereas pharmacological inhibition of γ-secretase has been linked to cognitive worsening in AD affected individuals (*Doody et al., 2013*). Here, we found *in vitro* that human (not murine) Aβ42 can inhibit γ-secretase processing, resulting in (i) the accumulation of substrates at the membrane, (ii) reductions in the release of soluble intracellular fragments from substrates, and (iii) dysregulation of γ-secretase dependent signaling.

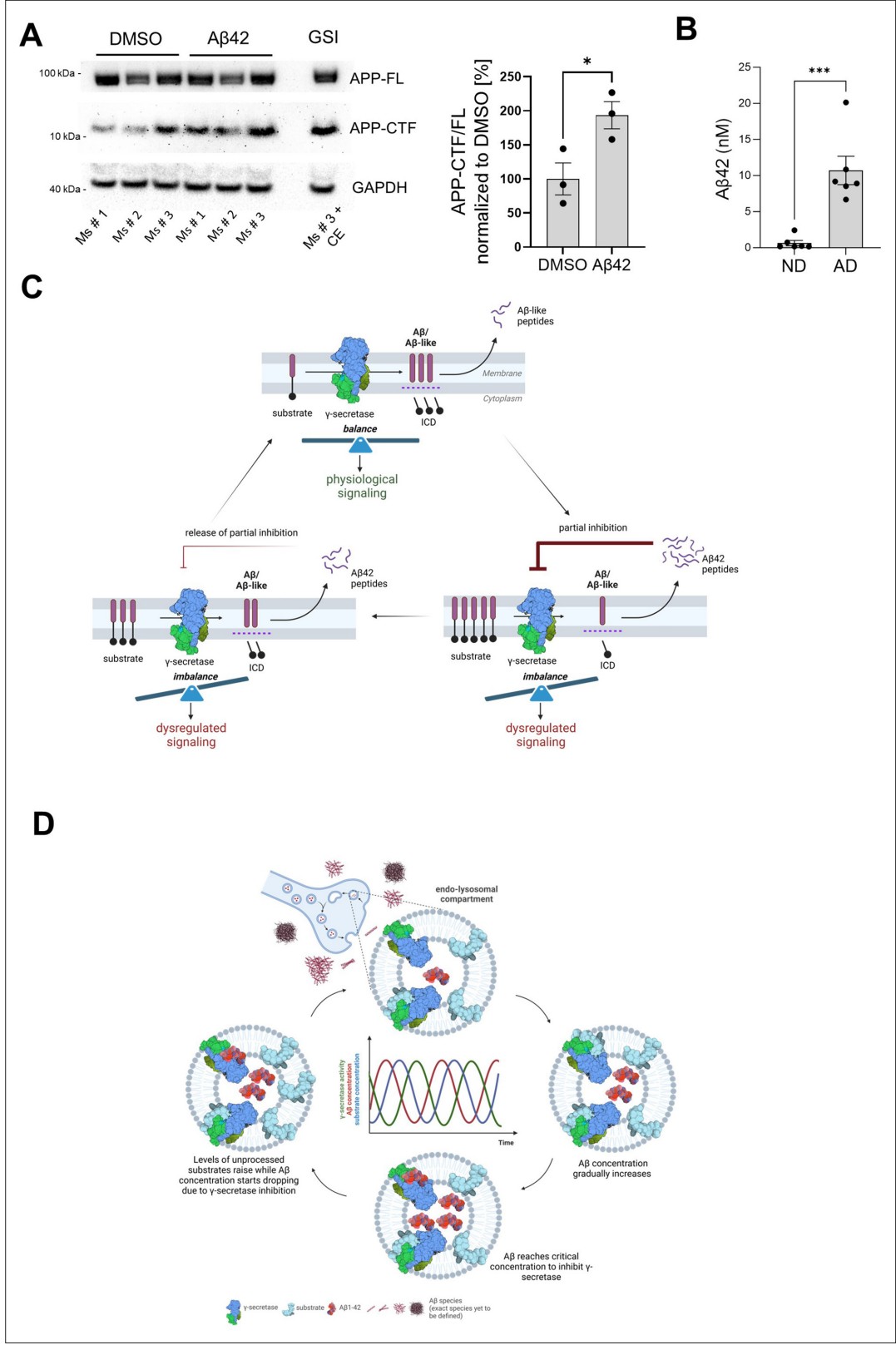

**Figure 10.** Aβ42 accumulates in Alzheimer's disease (AD)-linked synaptosomes and treatment with this peptide increases APP-CTFs in control, mouse-derived synaptosomes. (**A**) Synaptosomes from brain cortices of three wild-type mice were isolated, treated with DMSO or Aβ1–42, and analyzed by western blotting for APP-FL and APP-CTFs. Note the increase in APP-CTF/ FL ratio in Aβ1–42 treated samples relative to the control. The data are

*Figure 10 continued on next page*

*Figure 10 continued*

presented as mean ± SEM, n=3. The statistics were calculated using an unpaired Student's t-test. *p<0.05. (**B**) Aβ42 levels in synaptosomes derived from frontal cortices of post-mortem AD and age-matched non-demented (ND) control individuals were measured by ELISA. The data are presented as mean ± SEM, n=3. The statistics were calculated using an unpaired Student's t-test. ***p<0.001. (**C**) Schematic representation of the amyloid β (Aβ)-driven γ-secretase inhibitory feedback model is shown. (**D**) Scheme of the inhibitory model. Pathological increments in Aβ42 and endosomal accumulation of the peptide facilitate the establishment of an inhibitory product feedback mechanism that results in impairments in γ-secretase-mediated homeostatic signaling and contributes to AD progression. The inhibitory mechanism is complex, competitive, and reversible, and hence results in pulses of γ-secretase inhibition.

The online version of this article includes the following source data for figure 10:

**Source data 1.** The unedited blots and figures with the uncropped blots with the relevant bands clearly labeled.

Our analysis of the global γ-secretase (endopeptidase) activity in well-controlled cell-free assays, using purified protease and substrate, demonstrated that human Aβ42 inhibited the processing of APP$_{C99}$ by all members of the human γ-secretase family. Notably, despite the high homology between human and murine Aβ sequences, murine Aβ1–42 failed to inhibit the proteolysis of APP$_{C99}$, implying that structural determinants in the N-terminal domain of Aβ are involved in the inhibitory mechanism. Consistent with this view, N-terminally truncated Aβ peptides (Aβ11–42 and p3 17–42) demonstrated reduced or no inhibitory properties, compared to human Aβ42. The p3 peptides lack the 16 amino acids long, hydrophilic, and disordered N-terminal domain present in Aβ, but do contain the two aggregation-prone regions (16–21 and 29–42) required for the assembly of oligomers (*Festa, 2019*; *Dulin et al., 2008*) and fibrils (*Higgins et al., 1996*; *López de la Paz and Serrano, 2004*; *Kuhn et al., 2020*). Despite p3 peptides' aggregation-prone behavior (*Kuhn and Raskatov, 2020*) and their accumulation in AD brain (*Higgins et al., 1996*; *Saido et al., 1996*; *Lalowski et al., 1996*; *Gowing et al., 1994*), the neurotoxicity of these peptides is not well defined (*Walsh et al., 2002*). Moreover, our analyses in cell-free systems revealed that the C-terminus of Aβ also modulates its intrinsic inhibitory properties, with shorter Aβ1-x species (x=37, 38, 40) either not acting as inhibitors or inhibiting the protease to a lesser degree than human Aβ42. The reported lower affinities of shorter Aβ peptides towards γ-secretase may, at least partially, explain their decreased inhibitory potencies (*Szaruga et al., 2017*).

Our analysis on cultured neurons and neuron-like cells showed that extracellularly applied human Aβ42, but not p3, resulted in the accumulation of γ-secretase substrates. This phenotype is consistent with an Aβ42-induced inhibitory effect on γ-secretase activity, but could also be a consequence of alterations in cellular mechanisms that define protein (substrate) steady-state levels. However, Aβ42 treatment did not induce significant cellular toxicity, as determined by two independent readouts, nor did it promote APP-CTF generation, as indicated by the analysis of total, soluble APP ectodomain (precursor) levels. In addition, the assessment of the APP-CTF turnover by γ-secretase and/or general degradation mechanisms defined the inhibition of endogenous γ-secretases as the most important mechanism contributing to the accumulation of APP-CTF in living neurons treated with human Aβ42.

Intriguingly, despite their common inhibitory effects in cell-free conditions, only human Aβ42 and no other Aβ peptides induced the accumulation of unprocessed APP-CTFs in cells. Reports showing that Aβ conformation affects both cellular internalization and neurotoxicity (*Vadukul et al., 2020*) motivated us to test the possibility that Aβ42, unlike other Aβ peptides, acquires specific conformations that promote its endocytosis and/or inhibit its intracellular degradation. This would result in the intracellular concentration of this peptide at sites where γ-secretases process their substrates. Earlier reports of the selective uptake of Aβ42, relative to Aβ40 (*Hu et al., 2009*; *Wesén et al., 2017*; *Ling et al., 2014*; *Bahr et al., 1998*), facilitating the concentration of Aβ42 in spatially restricted endosomal compartments, would support this postulation. Our data that show marked increases in the intracellular levels of Aβ42, but neither Aβ40 nor Aβ43, add to the evidence for peptide-specific differences, and suggest that the observed increases mediate the inhibition of γ-secretases. Whether they are due to changes in Aβ42 cellular uptake, degradation rate or both requires further study.

Evidence that Aβ42 exerts a general inhibition on γ-secretase activity is provided by data showing that this peptide also reduced the processing of non-APP γ-secretase substrates, including NOTCH, ERBB4, neurexin, pan-cadherin, and p75-based substrates. Considering the poor sequence homology between distinct γ-secretase substrates and the general Aβ-mediated inhibitory effect, we reason that the inhibitory action of Aβ42 involves its interaction with the protease, rather than with the substrate.

Our analysis also showed that Aβ42-induced inhibition of γ-secretases alters downstream signaling events. Treatment of BFCNs (or PC12 cells) with human Aβ42, but not with p3, compromised the processing of p75, increased p75-CTF levels, and, in the absence of TrkA, induced apoptotic cell death. These observations recapitulated reported findings showing that GSI treatment of BFCNs triggers p75-driven cell death under similar conditions (*Franco et al., 2021*). It is noteworthy that these studies also showed that apoptosis in BFCNs subjected to GSI and TrkA signaling inhibitor treatments was rescued in p75 KO BFCNs, demonstrating that cell death is p75-dependent (*Franco et al., 2021*). The finding that Aβ42 treatment mimics GSI-driven p75-dependent apoptosis in BFCNs supports the inhibitory role of Aβ42 and links it to dysregulation of downstream signaling cascades. Our findings may also provide mechanistic bases for previous observations showing that injection of Aβ42 into the hippocampus of adult mice resulted in the degeneration of choline acetyltransferase (ChAT)-positive forebrain neurons in wild-type, but not p75-NTR-deficient mice, and revealed a correlation between Aβ42-driven neuronal death and the accumulation of p75-CTFs (*Sotthibundhu et al., 2008*; *Coulson et al., 2008*).

Aβ endocytic uptake has been shown to concentrate Aβ42 to µM levels in the endolysosomal compartments (*Hu et al., 2009*; *Schützmann et al., 2021*), where the high peptide concentration and low pH are proposed to facilitate Aβ aggregation and toxicity. We speculate that this mechanism could in addition promote the inhibition of γ-secretases. Given the reported role for APP$_{C99}$ in inducing dysregulation of the endolysosomal network, the potential Aβ42-mediated inhibition of γ-secretase processing of APP-CTFs could drive dysregulation of endolysosomal function leading to changes in synaptic and axonal signaling (*Xu et al., 2016*; *Kwart et al., 2019*; *Kim et al., 2016*; *Weissmiller et al., 2015*; *Sawa et al., 2022*; *Salehi et al., 2006*; *Jiang et al., 2019*). Equally intriguing is the possibility that the general inhibition of γ-secretase substrates by Aβ42 could contribute to neuroinflammation by modifying microglia biology (*Hou et al., 2023*) and neurodegeneration, as reported previously for the genetic inactivation of these enzymes (*Acx et al., 2017*; *Wines-Samuelson et al., 2010*; *Tabuchi et al., 2009*; *Saura et al., 2004*).

From a mechanistic standpoint, the competitive nature of the Aβ42-mediated inhibition implies that it is partial, reversible, and regulated by the relative concentrations of the Aβ42 peptide (inhibitor) and the endogenous substrates (*Figure 10C and D*). The model that we put forward is that cellular uptake, as well as endosomal production of Aβ, result in increased intracellular concentration of Aβ42, facilitating γ-secretase inhibition and leading to the buildup of APP-CTFs (and γ-secretase substrates in general). As Aβ42 levels fall, the augmented concentration of substrates shifts the equilibrium towards their processing and subsequent Aβ production. As Aβ42 levels rise again, the equilibrium is shifted back towards the inhibition. This cyclic inhibitory mechanism will translate into pulses of (partial) γ-secretase inhibition, which will alter γ-secretase mediated-signaling (arising from increased CTF levels at the membrane or decreased release of soluble intracellular domains from substrates). These alterations may affect the dynamics of systems oscillating in the brain, such as NOTCH signaling, implicated in memory formation, and potentially others (related to e.g. cadherins, p75, or neuregulins). It is worth noting that oscillations in γ-secretase activity induced by treatment with a γ-secretase inhibitor semagacestat have been proposed to have contributed to the cognitive alterations observed in semagacestat-treated patients in the failed Phase-3 IDENTITY clinical trial (*Doody et al., 2013*) and that semagacestat, like Aβ42, acts as a high affinity competitor of substrates (*Koch et al., 2023*).

The convergence of Aβ42 and tau at the synapse has been proposed to underlie synaptic dysfunction in AD (*McInnes et al., 2018*; *Ittner et al., 2010*; *Roberson et al., 2007*; *Spires-Jones and Hyman, 2014*), and recent assessment of APP-CTF levels in synaptosome-enriched fractions from healthy control, SAD, and FAD brains (temporal cortices) has shown that APP fragments concentrate at higher levels in the synapse in AD-affected than in control individuals (*Ferrer-Raventós, 2023*). Our analysis adds that endogenous Aβ42 concentrates in synaptosomes derived from end-stage AD brains to reach ~10 nM, a concentration that in CM from human neurons inhibits γ-secretase in PC12 cells (*Figure 7*). Furthermore, the restricted localization of Aβ in endolysosomal vesicles, within

synaptosomes, likely increases the local peptide concentration to the levels that inhibit γ-secretase-mediated processing of substrates in this compartment. In addition, we argue that the deposition of Aβ42 in plaques may be preceded by a critical increase in the levels of Aβ present in endosomes and the cyclical inhibition of γ-secretase activity that we propose. Under this view, reductions in γ-secretase activity may be a (transient) downstream consequence of increases in Aβ due to failed clearance, as represented by plaque deposition, contributing to AD pathogenesis.

The Aβ-mediated inhibition of γ-secretase may also help to explain the intriguing accumulation of APP-CTFs in the heterozygous FAD brain (*Pera et al., 2013*). In this regard, the direct quantification of γ-secretase activity in detergent-resistant fractions prepared from post-mortem brain samples of healthy controls and FAD-linked mutation carriers revealed similar overall γ-secretase activity levels, indicating that the wild-type (PSEN1 and PSEN2) γ-secretase complexes rescue any potential mutation-driven deficits in the processing of APP (*Szaruga et al., 2015*). Yet APP-CTFs have been reported to accumulate in the FAD brain (*Ferrer-Raventós, 2023*; *Pera et al., 2013*) and the accumulation of APP-CTFs appears to correlate with Aβ levels at the synapse. The inhibition of γ-secretase by Aβ42 could resolve the apparent conflict. Indeed, our data could reconcile these two seemingly exclusive hypotheses on the effects of FAD mutations in *PSEN1* on the development of AD by noting that: (1) there is a mutation-driven enhanced generation of Aβ42 within the endolysosomal network; (2) that through both endosomal production and endocytosis Aβ42 increases to a level within the endolysosomal network sufficient to inhibit the γ-secretase complex; and (3) that in the case of FAD mutations the isolation of the γ-secretase releases Aβ42, thus restoring wild-type enzyme activity (*Veugelen et al., 2016*; *Shen and Kelleher, 2007*). Thus, increased levels of endolysosomal Aβ42 with concurrent inhibition of γ-secretase may be responsible, at least in part, for the apparent γ-secretase loss-of-function phenotypes.

Collectively, our data raise the intriguing possibility that increases in Aβ42 in the AD brain, and in particular in the endolysosomal compartment, facilitate the establishment of an Aβ-driven inhibitory mechanism that contributes to neurotoxicity by impairing critical γ-secretase signaling functions. By mechanistically connecting elevated Aβ42 levels with the accumulation of multiple γ-secretase substrates, our observations integrate disparate views as to which pathways lead to neurodegeneration and offer a novel conceptual framework for investigating the molecular and cellular bases of AD pathogenesis.

## Materials and methods

**Key resources table**

| Reagent type (species) or resource | Designation | Source or reference | Identifiers | Additional information |
|---|---|---|---|---|
| Antibody | anti-FLAG M2 (Mouse monoclonal) | Merck | Cat# F3165 RRID:AB_259529 | WB(1:2000) |
| Antibody | anti-ADAM10 (Rabbit monoclonal) | Abcam | Cat# ab124695 RRID:AB_10972023 | WB(1:1000) |
| Antibody | anti-APP (Rabbit polyclonal) | gift from Prof. Wim Annaert (B63) | | WB(1:2000) |
| Antibody | anti-APP (Rabbit monoclonal) | Abcam | Cat# ab32136 RRID:AB_2289606 | WB(1:5000) |
| Antibody | anti-APP (Mouse monoclonal) | Thermo Fisher Scientific | Cat#14-9749-82 RRID:AB_2572978 | WB(1:250) |
| Antibody | anti-BACE1 (Rabbit monoclonal) | Abcam | Cat# ab183612 RRID:AB_3094757 | WB(1:1000) |
| Antibody | anti-pan-cadherin (Rabbit polyclonal) | Thermo Fisher Scientific | Cat# 71–7100 RRID:AB_2533992 | WB(1:200) |
| Antibody | anti-p75 NTR (Rabbit polyclonal) | Merck Millipore | Cat# 07–476 RRID:AB_310649 | WB(1:1000) |
| Antibody | anti-Aβ (Mouse monoclonal) | Biolegend | Cat# 800703 RRID:AB_662812 | ICC(1:1000) |

*Continued on next page*

*Continued*

| Reagent type (species) or resource | Designation | Source or reference | Identifiers | Additional information |
|---|---|---|---|---|
| Antibody | Anti-choline acetyltransferase (Goat polyclonal) | Merck Millipore | Cat# AB144P RRID:AB_2079751 | ICC(1:200) |
| Antibody | anti-cleaved caspase-3 (Rabbit polyclonal) | Cell Signaling | Cat# 9661 S | ICC (1:400) |
| Peptide, recombinant protein | Aβ peptides | rPeptide | Cat#: A-1001–1, A-1002–1, A-1078–1, A-1058–1-RPE, A-1063–1-RPE, A-1005–1-RPE, A-1008–1-RPE, A-1053–1-RPE | For Aβ42 the following lots were used: 4261242T, 06021342T and 02092242T. |
| Commercial assay or kit | Nano-Glo HiBiT Extracellular Detection System | Promega | Cat# N2420 | |
| Commercial assay or kit | LDH-Glo Cytotoxicity Assay | Promega | Cat#J2380 | |
| Commercial assay or kit | CellTiter-Glo 2.0 Assay | Promega | Cat#G9241 | |

## Cell lines

| Name | Species | Reference |
|---|---|---|
| SH-SY5Y | *Homo sapiens* | ATCC:CRL-2266 |
| PC12 | *Rattus norvegicus* | ATCC:CRL-1721 |
| ReNcell VM | *Homo sapiens* | Merck, Cat# SCC008 |
| HEK293T | *Homo sapiens* | ATCC:CRL-3216 |

## Chemicals, peptides, and antibodies

Aβ peptides were purchased from rPeptide, resuspended in DMSO at 500 µM, aliquoted into single use 10 µl aliquots and stored at –80 °C. For Aβ42 the following lots were used: 4261242T, 06021342T and 02092242T. Γ-secretase inhibitors (Inhibitor X (InhX, L-685,458), DAPT and compound E (CE)) were purchased from Bioconnect, Sigma-Aldrich and Millipore, respectively. TrkA inhibitor K252α, cycloheximide and Bafilomycin A1 were purchased from Sigma Aldrich. The following antibodies were used: mouse anti-FLAG M2 (Sigma-Aldrich, F3165), rabbit anti-ADAM10 antibody (EPR5622, Abcam, ab124695), rabbit anti-APP (gift from Prof. Wim Annaert (B63)), rabbit anti-APP (Y188, Abcam, ab32136), mouse anti-APP (22C11, Thermo Fisher Scientific, 14-9749-82), rabbit anti-BACE1 (EPR19523, Abcam, ab183612), rabbit anti-pan-cadherin (Thermo Fisher Scientific, 71–7100), anti-p75 NTR (Millipore, 07–476), anti-Aβ (clone 4G8, Biolegend), anti-choline acetyltransferase (Millipore, AB144P), anti-cleaved caspase-3 (Cell Signaling, 9661 S), HRP-conjugated goat anti-rabbit (BioRad), Alexa Fluor 790-conjugated goat anti-mouse (Thermo Fisher Scientific), Cy3-conjugated donkey anti-rabbit (Jackson ImmunoResearch Laboratories), Alexa Fluor Plus 488-conjugated donkey anti-mouse (Thermo Fisher Scientific), biotinylated rabbit anti-goat (Jackson ImmunoResearch Laboratories) and Cy2-conjugated streptavidin (Jackson Immunoresearch Laboratories).

## AAV production

Preparation of the AAV-hSyn1-C99 Y-T biosensor was performed as described previously (*Maesako et al., 2020*). Briefly, the cDNA of the C99 Y-T probe was subcloned into a pAAV2/8 vector containing human synapsin 1 promoter and WPRE sequences (*Maesako et al., 2017*). The packaging of viruses was performed at the University of Pennsylvania Gene Therapy Program vector core (Philadelphia, PA). The virus titer was 4.95E+13 GC/ml.

## Expression and purification of γ-secretase and γ-secretase substrates

High Five insect cells were transduced with baculoviruses carrying cDNA encoding all γ-secretase subunits (wild-type PSEN1 or PSEN2, NCSTN, APH1A or APH1B, PEN2) and 72 h later the cells were collected for protein purification, as described before (*Szaruga et al., 2017*).

Recombinant γ-secretase substrates were expressed and purified using a mammalian cell expression system or baculovirus-mediated expression system in High Five insect cells as before (*Szaruga et al., 2017*; *Franco et al., 2021*). COS cells were transfected with pSG5 plasmid encoding NOTCH1- or P75-based γ-secretase substrate, tagged at the C-terminus with 3xFLAG. High Five insect cells were transduced with baculoviruses carrying cDNA encoding APP$_{C99}$, ERBB4ΔECT, and neurexinΔECT-tagged with 3xFLAG-prescission protease (PPS) cleavage site-GFP tandem at the C-terminus. The purity of the protein was analyzed by SDS-PAGE and Coomassie staining (InstantBlue Protein Stain, Expedeon).

### Preparation of detergent-resistant membranes

CHAPSO detergent-resistant membranes (DRMs) were prepared from High Five insect cells overexpressing wild-type γ-secretase complexes containing NCSTN, APH1B, PSEN1 and PEN2 subunits, as reported before (*Szaruga et al., 2017*).

### Cell-free γ-secretase activity assays

Purified γ-secretases (~25 nM) were mixed with respective, purified 3xFLAG-tagged γ-secretase substrates (0.4 or 1.2 µM) in 150 mM NaCl, 25 mM PIPES, 0.25% CHAPSO, 0.03% DDM, 0.1% phosphatidylcholine buffer and the reactions were incubated for 40 min at 37°C. Non-incubated reactions or reactions supplemented with 10 µM InhX served as negative controls. In the experiments testing for the reversibility of the inhibition, γ-secretases were first immobilized on sepharose beads covalently coupled to anti-γ-secretase nanobody. Following the first reaction containing Aβ or GSI, the beads were washed 3 x for a total duration of 30 min with 150 mM NaCl, 25 mM PIPES, 0.25% CHAPSO buffer, and a fresh substrate was added. For γ-secretase activity assays in membrane-like conditions (25 µl total reaction volume), 6.25 µl DRMs, re-suspended in 20 mM PIPES, 250 mM sucrose, 1 M EGTA, pH7 at the concentration of 1 µg/µl, were mixed with 6.25 µl APP$_{C99}$-3xFLAG or Aβ1–42 substrate in 150 mM NaCl, 25 mM PIPES, 0.03% DDM, 2.5 µl 10xMBS, and H$_2$O supplemented with DDM and CHAPSO, to achieve final concentration of the detergents 0.03% and 0.1%, respectively, and final concentration of the substrates 1.5 µM and 10 µM, respectively. The reaction mixes were incubated for 40 min at 37°C. Reactions supplemented with 10 µM InhX served as negative controls.

### Analysis of ICD generation in cell-free γ-secretase activity assays

*De novo* ICD generation was determined by western blotting. Briefly, the reaction mixtures were subjected to methanol:chloroform (1:2 v/v) extraction to remove the excess of unprocessed substrates, and the upper fractions (containing mainly the generated ICDs) were subjected to SDS-PAGE. Otherwise, the high levels of substrate could preclude quantitative analysis of ICDs (*Figure 1—figure supplement 1A*).

ICD-3xFLAG was detected by western blotting with anti-FLAG, followed by anti-mouse Alexa Fluor 790 conjugated antibodies. Fluorescent signals were developed using Typhoon (GE Healthcare). The intensity of the bands was quantified by densitometry using ImageQuant software.

Alternatively, MALDI-TOF MS was applied to determine the relative *de novo* ICD-3xFLAG levels generated in cell-free conditions, as reported before (*Szaruga et al., 2017*). The samples were spiked with Aβ1–28 as an internal standard. The mass spectra were acquired on a RapiFleX TOF mass spectrometer (Bruker Daltonics) equipped with a 10 kHz Smartbeam laser using the AutoXecute function of the FlexControl 4.2 acquisition software.

### Analysis of γ-secretase substrate proteolysis in cultured cells using secreted HiBiT-Aβ or -Aβ-like peptide levels as a proxy for the global γ-secretase endopeptidase activity

HEK293 stably expressing APP-CTF (C99) or a NOTCH1-based substrate (similar in size as APP-C99) both N-terminally tagged with the HiBiT tag were plated at the density of 10,000 cells per 96-well, and 24 h after plating treated with Aβ or p3 peptides diluted in OPTIMEM (Thermo Fisher Scientific) supplemented with 5% FBS (Gibco). Conditioned media was collected and subjected to analysis using Nano-Glo HiBiT Extracellular Detection System (Promega). Briefly, 50 µl of the medium was mixed with 50 µl of the reaction mixture containing LgBiT Protein (1:100) and Nano-Glo HiBiT Extracellular Substrate (1:50) in Nano-Glo HiBiT Extracellular Buffer, and the reaction was incubated for 10 min at

room temperature. Luminescence signal corresponding to the amount of the extracellular HiBiT-Aβ or -Aβ-like peptides was measured using a victor plate reader with default luminescence measurement settings.

## Analysis of γ-secretase substrate levels in cultured cells

SH-SY5Y, PC12 and PC12nnr5 (*Loeb et al., 1991*) cell lines were cultured in Dulbecco's Modified Eagle Medium (DMEM)/F-12 (Thermo Fisher Scientific) supplemented with 10% fetal bovine serum (FBS) (Gibco) at 37°C, 5% $CO_2$. ReNCell VM cells were cultured in Corning Matrigel hESC-Qualified Matrix, LDEV-free-coated flasks in DMEM/F-12 medium supplemented with B27 (Thermo Fisher Scientific), 2 μg/ml heparin (Stem Cell Technologies), 20 ng/ml EGF-1 (Cell Guidance) and 25 ng/ml FGF (Cell Guidance). Human iPSC line was derived from a cognitively unaffected male individual, previously established and characterized (CVB) (RRID: CVCL_1N86, GM25430) (*Gore et al., 2011*). The two neuronal progenitor cell (NPC) lines – CV4a (derived from CVB iPSC line that carries a wild-type APP, herein termed as WT) and APPSwe (homozygous for APP Swedish mutations that were genome-edited from CVB parental iPSC line) were characterized and reported (*Young et al., 2015*; *Fong et al., 2018*). NPCs were plated on 20 μg/ml poly-L-ornithine (Sigma Aldrich) and 5 μg/ml laminin (Thermo Fisher Scientific)-coated plates in NPC media: DMEM/F12/Glutamax supplemented with N2, B27, penicillin, and streptomycin (Thermo Fisher Scientific) and 20 ng/ml FGF-2 (Millipore). For neuronal differentiation, confluent NPCs were cultured in NPC media without FGF-2 for three weeks. All cells were maintained in a humidified incubator at 37 °C, 5% $CO_2$, and regularly tested for mycoplasma infection.

For the cellular assays, cells were plated into 6-well plates at the density of 300,000 cells per well and 24 h after plating treated with Aβ or p3 peptides diluted in OPTIMEM (Thermo Fisher Scientific) supplemented with 5% FBS (Gibco). In the experiments set to analyze sAPP in the conditioned medium, FBS was replaced with 1% knock-out serum replacement (KOSR) (Thermo Fisher Scientific) due to the interference of the antibodies present in the serum with western blotting. After 8 h, the medium was refreshed, using peptide-supplemented media, and the cultures were incubated at 37°C for 16 hr. Cell lysates were prepared in radioimmunoprecipitation (RIPA) buffer (25 mM Tris, 150 mM NaCl, 1% Nonidet P-40, 0.5% sodium deoxycholate, 0.1% SDS, pH 7.4) supplemented with protease inhibitors. Equal amounts of protein or conditioned medium were subjected to SDS-PAGE on 4–12% Bis-Tris gels in MES or MOPS buffer or 4–16% Tris-Tricine gels (*Das et al., 2013*) and western blot analysis. Chemiluminescent signals were developed using Fujifilm LAS-3000 Imager or ChemiDoc XRS + imaging apparatus (BioRad). The optical density of the bands was quantified using ImageQuant or Image J software.

## Cytotoxicity assays

Cytotoxicity was assessed using LDH-Glo Cytotoxicity Assay (Promega), which is a bioluminescent plate-based assay for quantifying LDH release into the culture medium upon plasma membrane damage. The assay was performed following the manufacturer's recommendation. Briefly, the culture medium was collected upon respective treatment and diluted 1 in 100 in LDH storage buffer (200 mM Tris-HCl (pH 7.3), 10% glycerol, 1% BSA). Medium from cells treated for 15 min with 2% Triton X-100 was used as a maximum LDH release control. 12.5 μl of the diluted medium was mixed with 12.5 μl LDH detection enzyme mix supplemented with reductase substrate in a 384-well plate. The reactions were incubated for 1 h and luminescence read using Promega GloMax Discoverer.

Cytotoxicity was also assessed using CellTiter-Glo 2.0 Assay (Promega), which is a homogeneous method used to determine the number of viable cells based on the quantitation of the ATP, an indicator of metabolically active cells. Briefly, cells were plated in a 96-well plate in 100 μl medium, treated following the treatment regime, mixed with CellTiter-Glo reagent, and luminescence was read using Promega GloMax Discoverer.

## Isolation of biologically derived human Aβ and cellular assay for γ-secretase-mediated proteolysis

The conditioned media from the WT (WT-CM) and APPSwe (SWE-CM) human neurons were used as a biological source of Aβ. Following three weeks of differentiation, neurons cultured on 10 cm diameter dishes were incubated in NPC media without FGF-2 for five days. After five days, conditioned

media were collected in a sterile atmosphere. Immunodepletion of Aβ from APPSwe medium was done following a modified protocol (*Shankar et al., 2011*). Briefly, 5 ml of SWE-CM were incubated with 3D6 antibody (1 µg/ml), 15 µl protein A, and 15 µl protein G resin for 3 hr at 4°C with gentle shaking on a nutator (*Johnson-Wood et al., 1997*). The supernatant was collected after centrifuging at 3500×g and used as immunodepleted media (delSWE-CM) for cell culture experiments. To test for the APP-CTF accumulation in the presence of different conditioned media, PC12 cells, cultured on six-well plates, were treated with WT-CM, SWE-CM, or delSWE-CM for 24 h along with regular feeding media (control). Following incubation, cells were harvested and processed to detect APP-FL and APP-CTF.

## Spectral FRET analysis of γ-secretase activity in living neurons

Primary neuronal cultures were obtained from the cerebral cortex of mouse embryos at gestation days 14–16 (Charles River Laboratories). The neurons were dissociated using the Papain Dissociation System (Worthington Biochemical Corporation, Lakewood, NJ) and maintained for 13–15 days *in vitro* (DIV) in a neurobasal medium containing 2% B27, 1% GlutaMAX Supplement and 1% penicillin/streptomycin (Thermo Fisher Scientific). A laser at 405 nm wavelength was used to excite TurquoiseGL in the C99 Y-T biosensor (*Maesako et al., 2020*). The emitted fluorescence from the donors (TurquoiseGL) and the acceptors (YPet) was detected at 470±10 nm and 530±10 nm, respectively, by the TruSpectral detectors on an Olympus FV3000RS confocal microscope equipped with $CO_2$/heating units. 10/0.25x objective was used for the imaging. The Average pixel fluorescence intensity for the cell body after subtraction of the background fluorescence was measured using Image J. The emission intensity of YPet over that of the TurquoiseGL (Y/T) ratio was used as a readout of the FRET efficiency, which reflects the relative proximity between the donor and the acceptor. Pseudo-colored images were generated in MATLAB (MathWorks, Natick, MA). The neuronal preparation procedure followed the NIH guidelines for the use of animals in experiments and was approved by the Massachusetts General Hospital Animal Care and Use Committee (2003N000243).

## Primary cultures of basal forebrain cholinergic neurons

Animals of CD1 genetic background were housed in an animal care facility with a 12 h dark/light cycle and had free access to food and water. All experiments were performed according to the animal care guidelines of the European Community Council (86 / 609 / EEC) and to Spanish regulations (RD1201 / 2005), following protocols approved by the ethics committees of the Consejo Superior Investigaciones Científicas (CSIC). Septal areas were dissected from CD1 E17-18 embryos in chilled Hanks' balanced salt solution (HBSS, 137 mM NaCl, 5.4 mM KCl, 0.17 mM $Na_2HPO_4$, 0.22 mM $KH_2PO_4$, 9.9 mM HEPES, 8.3 mM glucose, and 11 mM sucrose). The septa were pooled and digested with 1 ml of 0.25% trypsin (GE Healthcare) and 0.5 ml of 100 kU DNase I (GE healthcare) for 10 min at 37°C. The tissue was further dissociated by aspiration with progressively narrower tips in neurobasal medium (Thermo Fisher Scientific) supplemented with 4% bovine serum albumin (BSA), 2% B27 (Thermo Fisher Scientific), 1% L-glutamine (Thermo Fisher Scientific) and 0.5% penicillin/streptomycin (GE Healthcare) (NB/B27). After tissue dissociation, an equal volume of 4% BSA was added and samples were centrifuged at 300×g for 5 min at 4 °C. The supernatant was discarded and the cell pellet was resuspended in 5 ml of 0.2% BSA. The suspension was filtered through a 40 µm nylon filter (Sysmex) and cells were counted in the Neubauer chamber, centrifuged again and resuspended in NB/B27. Cells were seeded in 24-well plates ($2\times10^5$ cells/well) containing 12 mm diameter pre-coated coverslips (VWR) for immunocytochemical analysis or in pre-coated p-100 plates for western blotting. The surfaces were coated with 50 µg/ml poly-D-lysine (Sigma-Aldrich) overnight at 4°C and 5 µg/ml laminin (Sigma-Aldrich) for 2 h at 37 °C. The next day, half of the medium was exchanged and the concentration of B27 was reduced to 0.2%. The medium was supplemented with 2 µM of the antimitotic agent (1-β-D-Arabinofuranosylcytosine (Sigma Aldrich)) and 100 ng/ml NGF (Alomone labs). Neurons were cultured until DIV11. The respective treatments were applied at DIV8 and the culture continued for 72 h until DIV11.

## Immunocytochemical analysis of the cell death rate

For immunocytochemistry (ICC) cells were fixed in 2% paraformaldehyde and permeabilized with 0.1% Triton X-100. After that, they were incubated with 0.5% SDS for 5 min for antigen retrieval.

Non-specific binding of antibodies was blocked using 2% BSA, 0.1% Triton X-100 in 0.1 M phosphate buffer (PB). The cells were incubated overnight with anti-choline acetyltransferase (Millipore) (only for BFCN) and anti-cleaved caspase-3 (Cell Signaling) antibodies at 4°C. The next day the samples were incubated with corresponding secondary antibodies: Cy3-conjugated donkey anti-rabbit, biotinylated rabbit anti-goat and Cy2-conjugated streptavidin (all purchased from Jackson Immunoresearch Laboratories). Nuclei were stained with DAPI and samples were mounted on glass slides. For quantitative analysis, for BFCNs total number of choline acetyltransferase positive neurons and the number of double positive neurons for choline acetyltransferase and cleaved caspase-3 was determined, while for PC12 a fraction of cleaved caspase 3 positive cells among all cells (determined by DAPI staining) was determined.

### Immunocytochemical analysis of Aβ internalization

PC12 cells were plated on glass coverslips and treated for 24 h with respective Aβ peptides at 1 μM. The cells were fixed with 4% paraformaldehyde and permeabilized with 0.5% Triton X-100 in TBST (0.2% Tween 20 in TBS). Blocking was performed by 1 h incubation in a solution containing 1% bovine serum albumin, 0.1% gelatin, 300 mM glycine, and 4% normal donkey serum in TBST. Primary antibodies were diluted in the blocking solution and incubated overnight. This was followed by incubation with respective Alexa Fluor Plus 488 conjugated secondary antibody and Alexa Fluor Plus 555 conjugated phalloidin. The samples were then incubated with nuclear stain DAPI for 5 min and mounted using Prolong glass. Images were acquired at 63x magnification using a Zeiss LSM 900 confocal microscope at VIB Bioimaging Core.

### Measurement of Aβ42 levels in synaptosomes from post-mortem human cortical tissue

The *post-mortem* human brain tissues from the frontal cortex were obtained from the University of California San Diego–Alzheimer's Disease Research Center (UCSD-ADRC; La Jolla, CA, USA), and the University of California Irvine–Alzheimer's Disease Research Center (UCI-ADRC) brain tissue repository (Irvine, CA, USA; *Supplementary file 1b*), and stored at -80 °C prior to use. Synaptosomes from the cortices were isolated following published protocol using a discontinuous Percoll gradient (Sigma-Aldrich, USA) (*Dunkley et al., 2008*; *Fonseca-Ornelas et al., 2021*) with some modifications. All the steps were performed on ice or at 4°C unless mentioned otherwise. Briefly, 1200 mg of tissue was homogenized using a handhold Dounce homogenizer in five volumes of homogenization buffer (HB: 0.32 M sucrose, 5 mM magnesium chloride, 5 mM Tris-Cl, pH 7.4 and EDTA-free protease inhibitor cocktail). After centrifuging at 1000xg for 10 min, post-nuclear supernatant (PNS) was collected. The PNS was further centrifuged at 10800×g for 10 min to isolate the pellet fractions (P2, crude synaptosome). The P2 fraction was resuspended in HB and loaded on top of a three-step (3%, 10%, and 23%) Percoll gradient, and centrifuged at 14,500×g for 12 min at 4°C in a Fiberlite F21-8×50 y Fixed-Angle Rotor (Thermo Fisher, USA). The synaptosome-rich interface between 10% and 23% Percoll layers was collected and resuspended in 30 volumes of HB. The diluted material was centrifuged at 18,500×g for 30 min at 4°C, and the synaptosome-enriched pellet was resuspended in 500 μl of HB. To measure the concentration of Aβ42, 25 μl of synaptosome suspension was centrifuged at 18500xg for 10 min at 4°C. Supernatant was discarded and the volume of the synaptosome pellet was estimated by adding a minimum volume of liquid (2 μl of PBS) to the synaptosome pellet. Synaptosomes were dissolved in 23 μl RIPA buffer (25 mM Tris-HCl pH 7.5, 150 mM NaCl, 1% NP-40, 1% sodium deoxycholate, 0.1% SDS) containing Halt Protease and Phosphatase Inhibitor Cocktail (Thermo Fisher Scientific). After centrifugation for 15 min at 13,800×g at 4°C, the supernatant was collected for MSD-ELISA (Meso Scale Diagnostics, USA). Concentrations of Aβ were measured using the VPLEXAβ Peptide Panel (4G8: MesoScaleDiscovery) following the manufacturer's Instructions. Data were obtained using the MESO QuickPlex SQ 120 and analyzed using DISCOVERY WORKBENCH 3.0 (Meso Scale Diagnostics). The pg concentration of Aβ in 25 μL was converted to pg/mL and eventually to nM values using the volume of synaptosomes.

## Measurement of APP-CTF levels in peptide-treated synaptosomes from mouse cortical tissue

The cortical tissues from two-months old wild-type mice (n=3) were collected and stored at -80 °C. Synaptosomes from the 60–70 mg of cortices were isolated following the protocol described above. The synaptosome-rich interface between 10% and 23% Percoll layers were collected and resuspended in 30 volumes of HB. The diluted material was centrifuged at 18500×g for 30 min at 4°C, and the synaptosome-enriched pellet was resuspended in HB supplemented with 10 mM glucose. 10–15 µg of synaptosome was incubated with Aβ42 peptide at 2.5 µM final concentration at 37 °C for 18 hr. DMSO was used as a vehicle control. One synaptosomal sample was treated with 200 nM of Compound E. Following incubation, samples were resolved on SDS-PAGE, and western blotting was performed using anti-APP Y188 and anti-GAPDH antibodies. All densitometric analyses were performed using NIH ImageJ software. The animal experiments were approved by the Institutional Animal Care and Use Committee of the University of California San Diego.

## Statistics

Statistical analysis was performed using Excel, GraphPad Prism, R 4.2.2. and R Studio software. The following R packages were used for the analysis: readxl, ggplot2, plyr, dplyr, DescTools, gridExtra and reshape2 (*Wickham, 2016*; *Wickham, 2007*; *Wickham, 2011*). $p < 0.05$ was considered as a predetermined threshold for statistical significance. One-way or two-way ANOVA, or Kruskal-Wallis test followed by Dunnett's, Tukey's, or Dunn multiple comparison test or unpaired Student's t-test were used, as described in the legends.

## Acknowledgements

This work was supported by the Cure Alzheimer's Fund (WCM-LCG), the Research Foundation Flanders (FWO, Research project G0B2519N) (LCG), the DH Chen Foundation (WCM), NIH AG015379 (OB), NIH AG044486 (OB), NIH R01AG055523 (WCM) and Bundesministerium für Bildung und Forschung (BMBF) grant M²OGA within the Partnership for Innovation in Health Industry, M²Aind (CH). MV acknowledges the Spanish Ministry of Science and Innovation (grant PID2021-127600NB-I00). We would like to thank VIB Bioimaging Core for support with confocal imaging, VIB Flow Core (Dr Jochen Lamote and Gonzalo Delgado Martinez) for support with FACS, and Dr Laetitia Miguel for support in the culture of ReNcell VM cells. We thank all the Mobley and Chávez-Gutiérrez lab members for fruitful discussions.

## Additional information

### Competing interests

William Mobley: consulted for Samumed and AC immune. He served on advisory boards for the Bluefield Project to Cure Frontotemporal Dementia, the Blythedale-Burke Pediatric Neuroscience Research Collaboration, The Key, the National Down Syndrome Society, the American Neurological Association, the Sanford Health Lorraine Cross Award Committee, the NIH COBRE program at the University of Nebraska, the Dementia Aware Committee and the Dementia Committee for the State of California Health Services, and the San Diego Alzheimer's Project. He was a member of a Pfizer Data Safety Monitoring Board. WM is a co-inventor on UCSD Patents for Gamma-secretase Modulators. WM received grant or contract funding from the NIH, Ono Pharma Foundation, Cure Alzheimer Fund, DH Chen Foundation, AC Immune, Larry L Hillblom Foundation, Alzheimer Association, Annovis-Bio and BioSplice and the Michael J Fox Foundation. He received a travel reimbursement from AC Immune. Annovis Bio provided a gift to the WM lab and a test compound. The other authors declare that no competing interests exist.

## Funding

| Funder | Grant reference number | Author |
| --- | --- | --- |
| Research Foundation Flanders | G0B2519N | Lucía Chávez-Gutiérrez |
| Cure Alzheimer's Fund | | William Mobley<br>Lucía Chávez-Gutiérrez |
| DH Chen Foundation | | William Mobley |
| National Institutes of Health | AG015379 | Oksana Berezovska |
| National Institutes of Health | AG044486 | Oksana Berezovska |
| National Institutes of Health | R01AG055523 | William Mobley |
| Bundesministerium für Bildung und Forschung | | Carsten Hopf |
| Spanish Ministry of Science and Innovation | PID2021-127600NB-I00 | Marçal Vilar |

The funders had no role in study design, data collection and interpretation, or the decision to submit the work for publication.

## Author contributions

Katarzyna Marta Zoltowska, Utpal Das, Conceptualization, Formal analysis, Visualization, Methodology, Writing – original draft, Writing – review and editing; Sam Lismont, Mei CQ Houser, Burcu Özcan, Diana Gomes Moreira, Dmitry Karachentsev, Ann Becker, Formal analysis, Methodology; Thomas Enzlein, Masato Maesako, Formal analysis, Visualization, Methodology, Writing – review and editing; Maria Luisa Franco, Formal analysis, Visualization, Methodology; Carsten Hopf, Marçal Vilar, Oksana Berezovska, Funding acquisition, Visualization, Writing – review and editing; William Mobley, Lucía Chávez-Gutiérrez, Conceptualization, Funding acquisition, Visualization, Writing – original draft, Writing – review and editing, Supervision

## Author ORCIDs

Katarzyna Marta Zoltowska ⓘ http://orcid.org/0000-0001-5853-3465
Masato Maesako ⓘ http://orcid.org/0000-0002-1970-2462
Burcu Özcan ⓘ http://orcid.org/0000-0001-9930-5274
Diana Gomes Moreira ⓘ http://orcid.org/0000-0003-2937-1772
Marçal Vilar ⓘ http://orcid.org/0000-0002-9376-6544
Oksana Berezovska ⓘ http://orcid.org/0000-0003-4898-5788
Lucía Chávez-Gutiérrez ⓘ http://orcid.org/0000-0002-8239-559X

## Ethics

For the experiments performed at Massachusetts General Hospital, the neuronal preparation procedure followed the NIH guidelines for the use of animals in experiments and was approved by the Massachusetts General Hospital Animal Care and Use Committee (2003N000243).For the experiments performed at Institute of Biomedicine of València, all procedures were performed according to the animal care guidelines of the European Community Council (86 / 609 / EEC) and to Spanish regulations (RD1201 / 2005), following protocols approved by the ethics committees of the Consejo Superior Investigaciones Científicas (CSIC). For the experiments performed at University of California San Diego, the procedures were approved by the Institutional Animal Care and Use Committee of the University of California San Diego.

Reviewer #1 (Public Review): https://doi.org/10.7554/eLife.90690.3.sa1
Author response https://doi.org/10.7554/eLife.90690.3.sa2

## Additional files

### Supplementary files
• Supplementary file 1. Supplementary tables included in the article. (a) Table summarizing IC50s in cell-free detergent-based γ-secretase activity assays for selected Amyloid β (Aβ) peptides. (b) Table summarizing demographics of the human post-mortem brain tissue.
• MDAR checklist

### Data availability
The authors confirm that the data supporting the findings of this study are available within the article and its supplementary materials.

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
