## [Editor Report · eLife assessment]

In this manuscript, the authors tested the hypothesis that Aβ42 toxicity arises from its proven affinity for γ-secretases. The authors provide **useful** findings, showing **convincingly** that human Abeta42 inhibits gamma-secretase activity. The data will be of interest to all scientists working on neurodegenerative diseases.

---

## [Referee Report · Reviewer #1 (Public Review)]

Summary:

Human Abeta42 inhibits gamma-secretase activity in biochemical assays.

Strengths:

Determination of inhibitory concentration human Abeta42 on gamma-secretase activity in biochemical assays.

---

## [Author Response]

The following is the authors’ response to the original reviews.

**Reviewer #1 (Recommendations For The Authors):**
Major concerns:(1) It is not clear about the biological significance of the inhibitory effects of human Abeta42 on gammasecretase activity. As the authors mentioned in the Discussion, it is plausible that Abeta42 may concentrate up to microM level in endosomes. However, subsets of FAD mutations in APP and presenilin 1 and 2 increase Abeta42/Abeta40 ratio and lead to Abeta42 deposition in brain. APP knock-in mice NLF and NLGF also develop Abeta42 deposition in age-dependent manner, although they produce more human Abeta42 than human Abeta40.If the production of Abeta42 is attenuated, which results in less Abeta42 deposition in brain. So, it is unlikely that human Abeta42 interferes gamma-secretase activity in physiological conditions. This reviewer has an impression that inhibition of gamma-secretase by human Abeta42 is an interesting artifact in high Abeta42 concentration. If the authors disagree with this reviewer's comment, this manuscript needs more discussion in this point of view.

We thank the Reviewer for raising this key conceptual point, we acknowledge that it was insufficiently discussed in the original manuscript. In response to this point, we introduced the following paragraph in the discussion section of the revised manuscript:

“From a mechanistic standpoint, the competitive nature of the Aβ42-mediated inhibition implies that it is partial, reversible, and regulated by the relative concentrations of the Aβ42 peptide (inhibitor) and the endogenous substrates (Figure 10C and 10D). The model that we put forward is that cellular uptake, as well as endosomal production of Aβ, result in increased intracellular concentration of Aβ42, facilitating γ-secretase inhibition and leading to the buildup of APP-CTFs (and γ-secretase substrates in general). As Aβ42 levels fall, the augmented concentration of substrates shifts the equilibrium towards their processing and subsequent Aβ production. As Aβ42 levels rise again, the equilibrium is shifted back towards inhibition. This cyclic inhibitory mechanism will translate into pulses of (partial) γsecretase inhibition, which will alter γ-secretase mediated-signaling (arising from increased CTF levels at the membrane or decreased release of soluble intracellular domains from substrates). These alterations may affect the dynamics of systems oscillating in the brain, such as NOTCH signaling, implicated in memory formation, and potentially others (related to e.g. cadherins, p75 or neuregulins). It is worth noting that oscillations in γ-secretase activity induced by treatment with a γ-secretase inhibitor semagacestat have been proposed to have contributed to the cognitive alterations observed in semagacestat treated patients in the failed Phase-3 IDENTITY clinical trial (*7*) and that semagacestat, like Aβ42, acts as a high affinity competitor of substrates (*85*).

The convergence of Aβ42 and tau at the synapse has been proposed to underlie synaptic dysfunction in AD (*86-89*), and recent assessment of APP-CTF levels in synaptosome-enriched fractions from healthy control, SAD and FAD brains (temporal cortices) has shown that APP fragments concentrate at higher levels in the synapse in AD-affected than in control individuals (*90*). Our analysis adds that endogenous Aβ42 concentrates in synaptosomes derived from end-stage AD brains to reach ~10 nM, a concentration that in CM from human neurons inhibits γ-secretase in PC12 cells (Figure 7). Furthermore, the restricted localization of Aβ in endolysosomal vesicles, within synaptosomes, likely increases the local peptide concentration to the levels that inhibit γ-secretase-mediated processing of substrates in this compartment. In addition, we argue that the deposition of Aβ42 in plaques may be preceded a critical increase in the levels of Aβ present in endosomes and the cyclical inhibition of γsecretase activity that we propose. Under this view, reductions in γ-secretase activity may be a (transient) downstream consequence of increases in Aβ due to failed clearance, as represented by plaque deposition, contributing to AD pathogenesis.”

We have also added figures 10C and 10D, presented here for convenience.

**Author response image 1. sa2fig1:** 

(2) It is not clear whether the FRET-based assay in living cells really reflects gamma-secretase activity.This reviewer thinks that the authors need at least biochemical data, such as levels of Abeta.

We have established a novel, HiBiT tag based assay reporting on the global γ-secretase activity in cells, using as a proxy the total levels of secreted HiBiT-tagged Aβ peptides. The assay and findings are presented in the revised manuscript as follows:

In the result section, in the “Aβ42 treatment leads to the accumulation of APP C-terminal fragments in neuronal cell lines and human neuron” subsection:

“The increments in the APP-CTF/FL ratio suggested that Aβ42 (partially) inhibits the global γ-secretase activity. To further investigate this, we measured the direct products of the γ-secretase mediated proteolysis of APP. Since the detection of the endogenous Aβ products via standard ELISA methods was precluded by the presence of exogenous human Aβ42 (treatment), we used an N-terminally tagged version of APPC99 and quantified the amount of total secreted Aβ, which is a proxy for the global γsecretase activity. Briefly, we overexpressed human APPC99 N-terminally tagged with a short 11 amino acid long HiBiT tag in human embryonic kidney (HEK) cells, treated these cultures with human Aβ42 or p3 17-42 peptides at 1 μM or DAPT (GSI) at 10 µM, and determined total HiBiT-Aβ levels in conditioned media (CM). DAPT was considered to result in full γ-secretase inhibition, and hence the values recorded in DAPT treated conditions were used for the background subtraction. We found a ~50% reduction in luminescence signal, directly linked to HiBiT-Aβ levels, in CM of cells treated with human Aβ42 and no effect of p3 peptide treatment, relative to the DMSO control (Figure 3D). The observed reduction in the total Aβ products is consistent with the partial inhibition of γ -secretase by Aβ42.”

In Methods:

“Analysis of γ-secretase substrate proteolysis in cultured cells using secreted HiBiT-Aβ or -Aβ-like peptide levels as a proxy for the global γ-secretase endopeptidase activity HEK293 stably expressing APP-CTF (C99) or a NOTCH1-based substrate (similar in size as APP- C99) both N-terminally tagged with the HiBiT tag were plated at the density of 10000 cells per 96-well, and 24h after plating treated with Aβ or p3 peptides diluted in OPTIMEM (Thermo Fisher Scientific) supplemented with 5% FBS (Gibco). Conditioned media was collected and subjected to analysis using Nano-Glo HiBiT Extracellular Detection System (Promega). Briefly, 50 µl of the medium was mixed with 50 µl of the reaction mixture containing LgBiT Protein (1:100) and Nano-Glo HiBiT Extracellular Substrate (1:50) in Nano-Glo HiBiT Extracellular Buffer, and the reaction was incubated for 10 minutes at room temperature. Luminescence signal corresponding to the amount of the extracellular HiBiT-Aβ or -Aβ-like peptides was measured using victor plate reader with default luminescence measurement settings.”

As the direct substrate of γ -secretase was used in this analysis, the observed reduction (~50%) in the levels of N-terminally-tagged (HiBiT) Aβ peptides in the presence of 1 µM Aβ42, relative to control conditions, demonstrates a selective inhibition of γ-secretase by Aβ42 (not by the p3). These data complement the FRET-based findings presented in Figure 5.

(3) Processing of APP-CTF in living cells is not only the cleavage by gamma-secretase. This reviewer thinks that the authors need at least biochemical data, such as levels of Abeta in Figures 4, 5 and 7.

We tried to measure the levels of Aβ peptides secreted by cells into the culture medium directly by ELISA (using different protocols) or MS (using established methods, as reported in Koch et al, 2023), but exogenous Aβ42 (treatment) present at relatively high levels interfered with the readout and rendered the analysis inconclusive.

However, we were successful in the determination of total secreted (HiBiT-tagged) Aβ peptides from the HiBiT tagged APP-C99 substrate, as indicated in the previous point. The quantification of the levels of these peptides showed that Aβ42 treatment resulted in ~50% reduction in the γ -secretase mediated processing of the tagged substrate.

In addition, we would like to highlight that our analysis of the contribution of other APP-CTF degradation pathways, using cycloheximide-based assays in the constant presence of γ-secretase inhibitor, failed to reveal significant differences between Aβ42 treated cells and controls (Figure 6B & C). The lack of a significant impact of Aβ42 on the half-life of APP-CTFs under the conditions of γsecretase inhibition maintained by inhibitor treatment is consistent with the proposed Aβ42-mediated inhibitory mechanism.

(4) Similar to comment #3. Processing of Pancad-CTF and p75 in living cells may be not only the cleavage by gamma-secretase. This reviewer thinks that the authors need at least biochemical data, such as levels of ICDs in Figures 6C and E.

To address this comment we have now performed additional experiments where we measured Nterminal Aβ-like peptides derived from NOTCH1-based substrate using the HiBiT-based assay. These experiments showed a reduction in the aforementioned peptides in the cells treated with Aβ42 relative to the vehicle control, and hence further confirmed the inhibitory action of Aβ42. These new data have been included as Figure 8D in the revised manuscript and described as follows:

Finally, we measured the direct N-terminal products generated by γ-secretase proteolysis from a HiBiT-tagged NOTCH1-based substrate, an estimate of the global γ-secretase activity. We quantified the Aβ-like peptides secreted by HEK 293 cells stably expressing this HiBiT-tagged substrate upon treatment with 1 µM Aβ1-42, p3 17-42 peptide or DAPT (GSI) (Figure 8D). DAPT treatment was considered to result in a complete γ-secretase inhibition, and hence the values recorded in the DAPT condition were used for background subtraction. A ~20% significant reduction in the amount of secreted N-terminal HiBiT-tagged peptides derived from the NOTCH1-based substrates in cells treated with Aβ1-42 supports the inhibitory action of Aβ1-42 on γ-secretase mediated proteolysis.

Minor concerns:(1) Murine Abeta42 may be converted to murine Abeta38 easily, compared to human Abeta42. This may be a reason why murine Abeta42 exhibits no inhibitory effect on gamma-secretase activity.

In order to address this question, we performed additional experiments where we assessed the processing of murine Aβ42 into Aβ38. Analogous to human Aβ42, the murine Aβ42 peptide was not processed to Aβ38 in the assay conditions. These new data have been integrated in the manuscript and added as a Supplementary figure 1B.

(2) It is curious to know the levels of C99 and C83 in cells in supplementary figure 3.

The conditions used in these assays were analogous to the conditions used in the figure 3 (i.e. treatment with Aβ peptides at 1 µM concentrations). Such conditions were associated with profound and consistent APP-CTF accumulation in this model system.

**Reviewer #2 (Recommendations For The Authors):**
In the current study, the authors show that Aβs with low affinity for γ-secretase, but when present at relatively high concentrations, can compete with the longer, higher affinity APPC99 substrate for binding and processing. They also performed kinetic analyses and demonstrate that human Aβ1-42 inhibits γ-secretase-mediated processing of APP C99 and other substrates. Interestingly, neither murine Aβ1-42 nor human p3 (17-42 amino acids in Aβ) peptides exerted inhibition under similar conditions. The authors also show that human Aβ1-42-mediated inhibition of γ-secretase activity results in the accumulation of unprocessed, which leads to p75-dependent activation of caspase 3 in basal forebrain cholinergic neurons (BFCNs) and PC12 cells.These analyses demonstrate that, as seen for γ-secretase inhibitors, Aβ1-42 potentiates this marker of apoptosis. However, these are no any in vivo data to support the physiological significance of the current finding. The author should show in APP KO mice whether gamma-secretase enzymatic activity is elevated or not, and putting back Aβ42 peptide will abolish these in vivo effects.

The findings presented in this manuscript form the basis for further *in vitro* and *in vivo* research to investigate the mechanisms of inhibition and its contribution to brain pathophysiology. Here, we used well-controlled model systems to investigate a novel mechanism of Aβ42 toxicity. Multiple mechanisms regulate the local concentration of Aβ42 *in vivo*, making the dissection of the biochemical mechanisms of the inhibition more complex. Nevertheless, beyond the scope of this report, we consider these very reasonable comments as a motivation for further research activities.

The experimental concentrations for Aβ42 peptide in the assay are too high, which are far beyond the physiological concentrations or pathological levels. The artificial observations are not supported by any in vivo experimental evidence.

It is correct that in the majority of the experiments we used low μM concentrations of Aβ42. However, we would like to note that we have also performed experiments where conditioned medium collected from human APP.Swe expressing neurons was used as a source of Aβ. In these experiments total Aβ concentration was in low nM range (0.5-1 nM) (Figure 7). Treatment with this conditioned medium led to the increase APP-CTF levels, supporting that low nM concentrations of Aβ are sufficient for partial inhibition of γ-secretase.

In addition, we highlight that analyses of the brains of the AD affected individuals have shown that APPCTFs accumulate in both sporadic and genetic forms of the disease (Pera et al. 2013, Vaillant-Beuchot et al. 2021); and recently, Ferrer-Raventós et al. 2023 have revealed a correlation between APP-CTFs and Aβ levels at the synapse (Ferrer-Raventós et al. 2023). We therefore assessed the concentration of Aβ42 in synaptosomes derived from frontal cortices of post-mortem AD and age-matched non-demented (ND) control individuals. Our findings and conclusions are included in the revised version as follows:

In the results section:

“We next investigated the levels of Aβ42 in synaptosomes derived from frontal cortices of post-mortem AD and age-matched non-demented (ND) control individuals (Figure 10B). Towards this, we prepared synaptosomes from frozen brain tissues using Percoll gradient procedure (*62, 63*). Intact synaptosomes were spun to obtain a pellet which was resuspended in minimum amount of PBS, allowing us to estimate the volume containing the resuspended synaptosome sample. This is likely an overestimate of the actual synaptosome volume. Finally, synaptosomes were lysed in RIPA buffer and Aβ peptide concentrations measured using ELISA (MSD). We observed that the concentration of Aβ42 in the synaptosomes from (end-stage) AD tissues was significantly higher (10.7 nM) than those isolated from non-demented tissues (0.7 nM), p<0.0005***. These data provide evidence for accumulation at nM concentrations of endogenous Aβ42 in synaptosomes in end-stage AD brains. Given that we measured Aβ42 concentration in synaptosomes, we speculate that even higher concentrations of this peptide may be present in the endolysosome vesicle system, and therein inhibit the endogenous processing of APP-CTF at the synapse. Of note treatment of PC12 cells with conditioned medium containing even lower amounts of Aβ (low nanomolar range (0.5-1 nM)) resulted in the accumulation of APP-CTFs.”

In the discussion:

“The convergence of Aβ42 and tau at the synapse has been proposed to underlie synaptic dysfunction in AD (*86-89*), and recent assessment of APP-CTF levels in synaptosome-enriched fractions from healthy control, SAD and FAD brains (temporal cortices) has shown that APP fragments concentrate at higher levels in the synapse in AD-affected than in control individuals (*90*). Our analysis adds that endogenous Aβ42 concentrates in synaptosomes derived from end-stage AD brains to reach ~10 nM, a concentration that in CM from human neurons inhibits γ-secretase in PC12 cells (Figure 7). Furthermore, the restricted localization of Aβ in endolysosomal vesicles, within synaptosomes, likely increases the local peptide concentration to the levels that inhibit γ-secretase-mediated processing of substrates in this compartment. In addition, we argue that the deposition of Aβ42 in plaques may be preceded by a critical increase in the levels of Aβ present in endosomes and the cyclical inhibition of γ-secretase activity that we propose. Under this view, reductions in γ-secretase activity may be a (transient) downstream consequence of increases in Aβ due to failed clearance, as represented by plaque deposition, contributing to AD pathogenesis. ”